# Coordinated control of terminal differentiation and restriction of cellular plasticity

**Tulsi Patel, Oliver Hobert\***

Department of Biological Sciences, Howard Hughes Medical Institute, Columbia University, New York, United States

**Abstract** The acquisition of a specific cellular identity is usually paralleled by a restriction of cellular plasticity. Whether and how these two processes are coordinated is poorly understood. Transcription factors called terminal selectors activate identity-specific effector genes during neuronal differentiation to define the structural and functional properties of a neuron. To study restriction of plasticity, we ectopically expressed *C. elegans* CHE-1, a terminal selector of ASE sensory neuron identity. In undifferentiated cells, ectopic expression of CHE-1 results in activation of ASE neuron type-specific effector genes. Once cells differentiate, their plasticity is restricted and ectopic expression of CHE-1 no longer results in activation of ASE effector genes. In striking contrast, removal of the respective terminal selectors of other sensory, inter-, or motor neuron types now enables ectopically expressed CHE-1 to activate its ASE-specific effector genes, indicating that terminal selectors not only activate effector gene batteries but also control the restriction of cellular plasticity. Terminal selectors mediate this restriction at least partially by organizing chromatin. The chromatin structure of a CHE-1 target locus is less compact in neurons that lack their resident terminal selector and genetic epistasis studies with H3K9 methyltransferases suggest that this chromatin modification acts downstream of a terminal selector to restrict plasticity. Taken together, terminal selectors activate identity-specific genes and make non-identity-defining genes less accessible, thereby serving as a checkpoint to coordinate identity specification with restriction of cellular plasticity.

\*For correspondence: or38@columbia.edu

## Introduction

The acquisition of differentiated cell identities in a multicellular organism is accompanied by the loss of developmental plasticity. Undifferentiated cells are pluripotent and have the potential to activate diverse categories of genes. As differentiation progresses, cells activate cellular identity-specific genes as well as repress non-identity-specific genes, thereby losing pluripotency and attaining a differentiated state that is restricted. In this work, we aimed to uncover molecules and mechanisms that coordinate identity specification with the restriction of cellular plasticity.

The progressive loss of developmental plasticity in differentiating cells was first observed in pioneering somatic nuclear transplant experiments (*Briggs and King, 1952*; *Gurdon, 1960*, *1962*). When nuclei from progressively more differentiated frog intestinal cells were transplanted into enucleated eggs, it was observed that egg cells containing less differentiated nuclei developed into tadpoles more efficiently than eggs containing more differentiated nuclei (*Gurdon, 1960*, *1962*). Similar trends have been noted in somatic nuclear transplant experiments performed since (*Gurdon and Wilmut, 2011*).

Experiments in which transcription factors (TFs) are ectopically expressed in progressively more differentiated cells also demonstrate restriction of plasticity. For instance, in *C. elegans*, the

overexpression of myogenic helix-loop-helix TF *hlh-1*/MyoD early in embryonic development leads to the activation of muscle genes in all somatic cells (*Fukushige and Krause, 2005*). However, the ability of *hlh-1* to activate its target genes decreases and is completely lost by the time the embryonic cells are post-mitotic. Similar observations are made when endodermal GATA TFs *end-1* and *elt-7* (*Zhu et al., 1998*), ectodermal GATA TFs *elt-1* and *elt-3* (*Gilleard and McGhee, 2001*; *Gilleard et al., 1999*; *Page et al., 1997*) and pharyngeal FoxA TF *pha-4* (*Horner et al., 1998*) are ectopically expressed in the worm embryo at different developmental time points. Likewise, the overexpression of pluripotency TFs c-Myc, Klf4, Sox2, and Oct4 in non-terminally differentiated Pro-B cells leads to the formation of iPSCs, but expression in differentiated B lymphocytes does not (*Hanna et al., 2008*). In all of these cases, restriction of plasticity is apparent in the inability of more differentiated cells to activate non-identity-specific genes in response to ectopically expressed TFs.

There is accumulating evidence that gene repression mediated by chromatin structure and organization plays an important role in restricting cellular plasticity (*Meister et al., 2011*). In *C. elegans* and mice, electron microscopy shows mostly open euchromatin in cells of early embryos and an accumulation of closed heterochromatic regions in differentiated cells (*Efroni et al., 2008*; *Leung et al., 1999*; *Park et al., 2004*). Differentiation is also coupled with changes in covalent histone modifications. These histone modifications, in conjunction with one another and their effector proteins can influence chromatin structure, nucleosome-DNA interactions, as well as accessibility to TFs (*Greer et al., 2014*; *Tessarz and Kouzarides, 2014*). The presence of activating and repressive chromatin marks can divide the genome of a differentiated cell into genes that can be expressed and not expressed, thereby contributing to restriction of plasticity (*Mohn and Schübeler, 2009*).

The repressive histone modifications H3K27me3 and H3K9me3 have been specifically implicated in the loss of plasticity. These marks are generally found in distinct chromatin regions in various species including humans, *Drosophila*, mice, and *C. elegans* (*Ernst and Kellis, 2010*; *Filion et al., 2010*; *Ho et al., 2014*). H3K27me3 is enriched on developmentally regulated, cell-type-specific genes (*Boyer et al., 2006*; *Filion et al., 2010*; *Ho et al., 2014*; *Oktaba et al., 2008*). A correlation between loss of H3K27me3, chromatin compaction, and plasticity has been shown in *C. elegans* embryos. The depletion of H3K27me3 in embryos leads to an extension of the time during which chromatin stays more 'open' and a correlated increase in the window of time during which an ectopically expressed TF, *pha-4*, can activate its target genes in embryonic cells (*Yuzyuk et al., 2009*). In previous studies, we have also shown that loss of H3K27me3 depositing polycomb repressive complex 2 (PRC2) in the germ cells of *C. elegans* leads to a gain of plasticity, as measured by the ability of three different identity-specifying TFs to activate their target genes (*Patel et al., 2012*). On the other hand, H3K9me3 is enriched at repeat-rich regions which remain repressed in all cell types, but there is accumulating evidence that it is also involved in the repression of cell-type-specific genes (*Allan et al., 2012*; *Feldman et al., 2006*; *Garrigues et al., 2015*; *Ho et al., 2014*; *Loh et al., 2007*; *Martens et al., 2005*; *Towbin et al., 2012*). As for restricting plasticity, it has been observed that removing methyl transferases that deposit H3K9me3 increases the efficiency of fibroblast to iPSC reprogramming (*Soufi et al., 2012*). Furthermore, H3K9me3 is enriched at targets of iPSC-inducing 'Yamanaka' factors that fail to become activated in fibroblasts that do not convert into iPSCs (*Soufi et al., 2012*; *Soufi and Zaret, 2013*).

During differentiation, cells activate identity-specific genes and form active and repressive chromatin landscapes (*Meister et al., 2010*; *Mohn and Schübeler, 2009*). How is the establishment of this chromatin-level organization coordinated with the transcription of identity-specific effector genes? Certain identity-specifying TFs facilitate the formation of active chromatin landscapes at *cis*-regulatory regions that they bind. The binding of these TFs leads to formation of more 'open' euchromatin by facilitating changes in histone modifications and chromatin accessibility (*Fakhouri et al., 2010*; *Iwafuchi-Doi and Zaret, 2014*; *Zaret and Carroll, 2011*, *2016*). These findings have led to the hypothesis that TFs that activate identity-specific genes might also function as organizers of identity-specific active chromatin (*Natoli, 2010*). TFs that regulate the formation of open chromatin landscapes are straight-forward to identify since their loss generally leads to the loss of cellular identity. Much less is known about the organization of heterochromatin that mediates restriction of plasticity. Genetic removal of heterochromatin organizers is not necessarily expected to lead to the derepression (i.e. activation) of numerous non-identity-specific genes under wildtype conditions. However, the loss of heterochromatin organizers might make genes more accessible, so that they can be activated if trans-activators are also present in the cell. For this reason,

the ectopic expression of TFs represents an experimental strategy for uncovering mechanisms that regulate heterochromatin organization and plasticity restriction.

In this work, we use a TF overexpression assay to show that cellular identity-specifying TFs also regulate the restriction of cellular plasticity at least partially by directing the formation of repressive chromatin in *C. elegans* neurons. In various animal species, it has been shown that the terminal differentiation of individual neuron types is carried out by one or more TFs called terminal selectors, which activate expression of a variety of identity-specific effector genes whose protein products shape the terminally differentiated state of a cell (*Hobert, 2011*, *2016*). In the absence of these TFs, neurons still migrate properly, extend processes, and acquire a 'generic' neuronal state characterized by the expression of pan-neuronal markers; but these neurons do not express neuron-type-specific genes such as ion channels, neurotransmitters, neurotransmitter receptors, etc. (*Hobert, 2011*; *Stefanakis et al., 2015*). We demonstrate here that terminal selector TFs also function to restrict cellular plasticity. To assay restriction, we have used the overexpression of terminal selector CHE-1, a Zn-finger TF, which is normally only expressed in two glutamatergic ASE sensory neurons and which specifies the identity of these neurons by activating expression of ASE-specific genes (*Etchberger et al., 2007*; *Patel et al., 2012*; *Serrano-Saiz et al., 2013*; *Uchida et al., 2003*). We show here that if CHE-1 is ectopically expressed during embryonic and early larval development, when cells are dividing and differentiating, it can activate expression of ASE-specific effector genes in many cell types. If CHE-1 is expressed at later stages when all cells have differentiated, it is no longer able to activate ASE effector genes in most cells, showing that they are restricted. We then expressed CHE-1 in wildtype and mutant animals in which terminal selectors of specific neuron types are genetically eliminated to assay whether the cells affected in the terminal selector mutants would remain responsive to CHE-1. We found this to be the case in five of seven terminal selector mutants tested, showing that several terminal selector TFs are required for the loss of plasticity.

To understand the mechanistic basis of the plasticity-restricting function of terminal selectors, we focused on *unc-3*, a COE (Collier/Olf1/Ebf) TF which specifies the identity of cholinergic motor neurons (MNs). We identified an allele of *unc-3* in which cholinergic MNs express identity-specific markers and function normally, yet are still plastic. This finding strongly suggests that even in an otherwise appropriately specified neuron, terminal selector function is required to restrict plasticity. Next, using a LacI/LacO spot assay (*Meister et al., 2010*; *Towbin et al., 2012*; *Yuzyuk et al., 2009*) to tag transgenic loci, we found that the locus of a reporter of a CHE-1 target gene is less condensed in *unc-3* mutant neurons compared to wildtype neurons, suggesting that terminal selector TFs are required for the formation of identity-specific heterochromatin. We also observe that removal of H3K27me and H3K9me depositing enzymes leads to loss of restriction in otherwise normally differentiated neurons, further emphasizing that chromatin does play a role in restricting plasticity of neurons. While the loss of H3K27 methyltransferase showed an additive effect with *unc-3*, the loss of H3K9 methyltransferases did not modify the *unc-3* phenotype suggesting that correct deposition of H3K9me3 may be downstream of *unc-3* function. Overall, this work demonstrates a novel role for terminal selector-like TFs in coordinating the processes of identity-specific gene activation with restriction of cellular plasticity, the latter being mediated at least partially through the correct organization of chromatin in differentiated cells.

## Results

### Differential ability of CHE-1 to activate target genes

In order to gain insights into the mechanisms that restrict plasticity, we probed cellular plasticity in many cell types of *C.elegans* throughout different developmental stages. To this end, we ubiquitously expressed Zn-finger TF CHE-1 at different time points in developing worms. CHE-1 is normally expressed only in the ASEs, two sensory neurons located in the head, and acts as a terminal selector by activating expression of numerous ASE-specific effector genes (*Chang et al., 2003*; *Etchberger et al., 2007*; *Rinn et al., 2007*; *Tursun et al., 2009*; *Uchida et al., 2003*). We used a previously described transgenic worm strain which contains a heatshock inducible promoter driving *che-1* expression (CHE-1[hs]), and a reporter of a CHE-1 target gene, *gcy-5*, that is normally expressed in ASER (*Patel et al., 2012*; *Tursun et al., 2011*). In this assay, non-ASE cells that activate expression of *gcy-5* in response to CHE-1[hs] expression are considered plastic, whereas cells that do not activate

*gcy-5* expression are considered restricted. In this study, we used several tools to detect the expression of *gcy-5*, including two kinds of transgenic reporters (*gcy-5$^{prom}$::gfp and gcy-5$^{fosmid}$::gfp*), smFISH probes against *gcy-5* mRNA, and an allele of *gcy-5* in which the endogenous locus is tagged with *mNeonGreen* using CRISPR/Cas9-mediated genome engineering (***Figure 1—figure supplement 1A,B,C***). *gcy-5$^{prom}$::gfp* uses 3.2 kb of regulatory sequences directly upstream of the start codon of the *gcy-5* gene, while the *gcy-5$^{fosmid}$::gfp* contains ~33 kb of DNA surrounding the *gcy-5* locus, which should include all potential *cis*-regulatory regions (***Figure 1—figure supplement 1A***). smFISH and the *mNeonGreen* tagged allele validate the endogenous transcription of *gcy-5*, but because the *mNeonGreen* tagged allele was difficult to detect by standard microscopy, we utilized smFISH for the detection of the endogenous transcript in our CHE-1 misexpression experiments. *gcy-5* is normally expressed in the ASER and RIGL/R neurons. Expression of *gcy-5* in the ASER is *che-1* dependent (***Chang et al., 2003***; ***Uchida et al., 2003***), while expression in the RIG neurons is dependent on *lim-6* (***Figure 1—figure supplement 1D***), and should not be relevant in CHE-1 over-expression assays.

When CHE-1$^{hs}$ is induced during early embryonic stages (<300 min), *gcy-5$^{prom}$::gfp* is broadly activated in numerous cells (***Figure 1Aiii***). However, these embryos become deformed and arrest, making it difficult to score exactly which and to what extent non-ASE cells are CHE-1-responsive. Induction of CHE-1$^{hs}$ during larval development results in an increasingly restricted activation of *gcy-5,* and the animals retain a normal overall morphology (***Figure 1Aiv–viii***). With the exception of a few neurons in the head and tail, cells that express *gcy-5* in response to larval CHE-1$^{hs}$ are those that have just been generated and hence are still differentiating (***Figure 1B***). For instance, CHE-1$^{hs}$ induction in the L2/L3 stage activates *gcy-5* in seam cells and ventral nerve cord (VNC) neurons, which are dividing and differentiating at this time. As some VNC neurons are born during embryonic stages, we examined whether VNC cells that activate *gcy-5* at L2 are those that are newly born. To this end, we used a pan-neuronal marker, *rab-3$^{prom}$::rfp*, which is brightly expressed in VNC neurons that were born during embryonic stages, but is dim in cells that are still undergoing differentiation. In these worms, activation of *gcy-5$^{fosmid}$::gfp* after L2 CHE-1$^{hs}$ expression is enriched in *rab-3$^{low}$* cells (***Figure 1—figure supplement 2A***), showing that cells that are dividing and differentiating are more receptive to CHE-1 than cells that have already differentiated. We also observe that vulval muscle cells, which are dividing and differentiating at the L3 stage are receptive to CHE-1$^{hs}$ at this stage (***Figure 1B***). At the L4 and adult stages, when all tissues have differentiated, the only cells that activate *gcy-5$^{prom}$::gfp* in response to CHE-1$^{hs}$ are a handful of neurons and pharyngeal muscle cells, suggesting that these differentiated cells provide some molecular context which makes them amenable to *che-1* activity (***Figure 1Aviii and B***). Such context-dependent activity has been previously observed for TF *elt-7*, which when overexpressed, leads to a context-dependent activation of endodermal identity in specific adult tissues (***Riddle et al., 2016***, ***2013***). The CHE-1 responsive neurons include the RIS, CEP, ASK, and ASI neurons. These neurons do not bear any obvious relationship to ASE, and it is unclear why these cells remain responsive to CHE-1.

The same pattern of *gcy-5$^{prom}$::gfp* induction is also seen when CHE$^{hs}$ is induced with a single copy *hs$^{prom}$::che-1* transgene generated by MiniMos-mediated genome engineering (***Frøkjær-Jensen et al., 2014***). Additionally, *gcy-5* smFISH of these worms shows that expression of *gcy-5$^{prom}$::gfp* overlaps with the expression of endogenous *gcy-5* mRNA, indicating that the endogenous *gcy-5* gene is induced by ectopic CHE-1$^{hs}$ and that the transgene *gcy-5$^{prom}$::gfp* reliably replicates the activity of the endogenous *gcy-5* locus (***Figure 1—figure supplement 2B***).

We confirmed that the restricted activation of *gcy-5* in adults is not due to an inability to efficiently induce CHE-1$^{hs}$. Ater heatshock induction in adults, both *che-1* mRNA and protein can be detected in all cell types (***Figure 1—figure supplement 2C,D***). Furthermore, to ensure that the expressed CHE-1 protein is functional, we generated a synthetic strong reporter that contains a 4X multimerized CHE-1 binding site ('ASE motif'), taken from the *gcy-5* locus, driving *gfp*. We find that this *gcy-5$^{4XASE}$::gfp* reporter is expressed broadly, in all somatic cell types at every stage of development, in response to CHE-1$^{hs}$ expression from both the multi-copy and single-copy heatshock-inducible transgenes (***Figure 1D***). Therefore, the restricted activation of *gcy-5* reflects the inability of CHE-1$^{hs}$ to activate its target genes in differentiated cells.

To determine the breadth of this phenomenon, we examined expression of two additional *che-1* target genes, *gcy-6 and gcy-7*, in the heatshock assay (***Chang et al., 2003***; ***Uchida et al., 2003***). These genes are receptor guanylate cyclases expressed exclusively in the ASEL neuron during

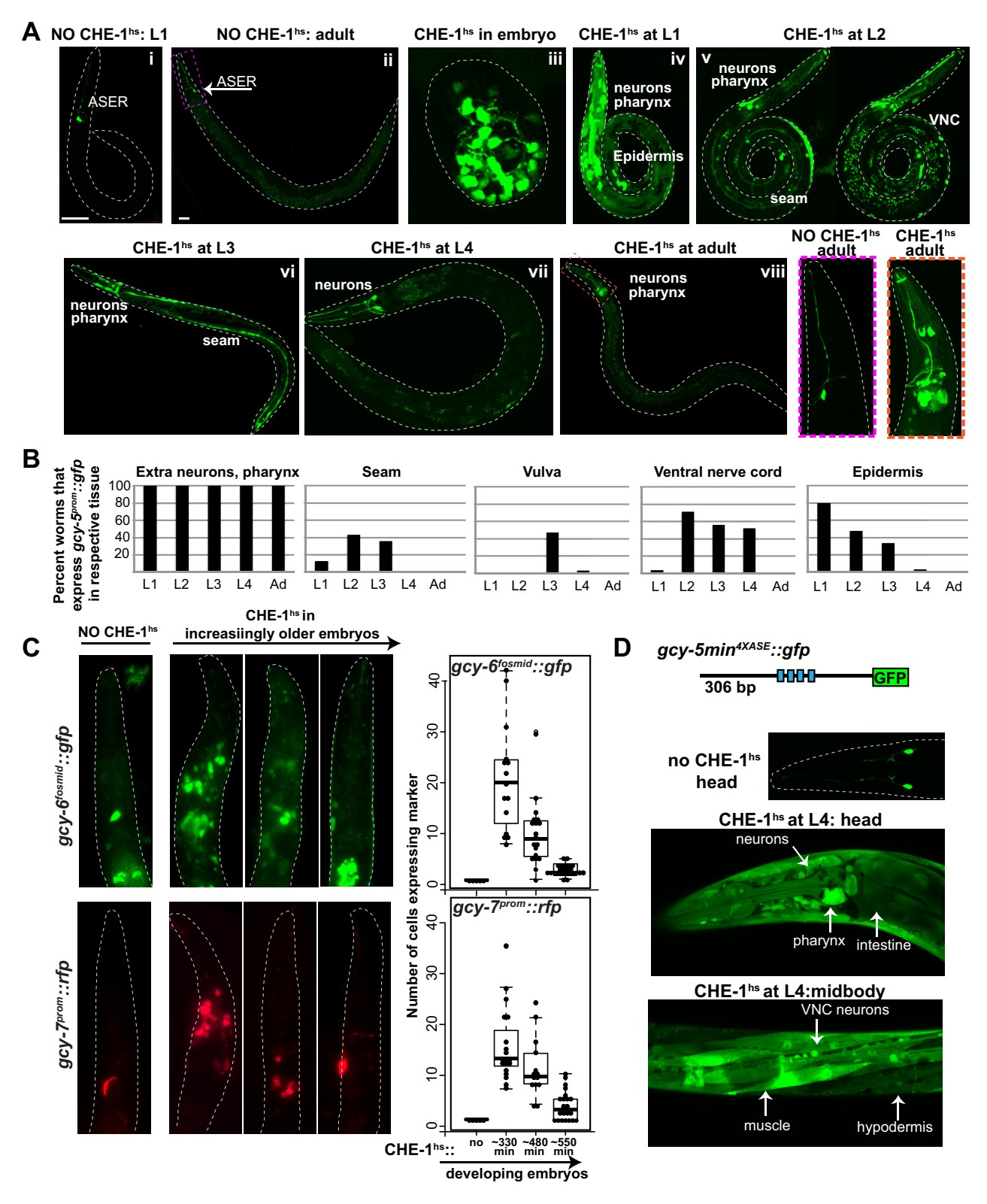

**Figure 1.** Expression of ASE markers in response to CHE-1hs at different developmental stages. (**A**) Expression of *gcy-5prom::gfp* becomes less broad as CHE-1hs is induced at later stages. (i) L1 worm without CHE-1hs, bright expression in ASER; (ii) adult without CHE-1hs, expression in ASER (arrow); (iii) embryonic CHE-1hs, broad *gcy-5prom::gfp* induction in deformed embryos; (iv) L1 CHE-1hs, expression of *gcy-5prom::gfp* in various tissues including hypodermis, various neurons, muscles, pharynx. (v) L2 CHE-1hs at two different focal points, *gcy-5prom::gfp* in seam, muscles, ventral nerve cord, other

*Figure 1 continued on next page*

*Figure 1 continued*

neurons and pharynx; (vi) L3 CHE-1$^{hs}$, gcy-5$^{prom}$::gfp in seam, vulva cells, neurons and pharynx; (vii) L4 and (viii) adult CHE-1$^{hs}$, gcy-5$^{prom}$::gfp is seen only in neurons and pharynx. Higher magnification of the heads in (i) and (viii). (B) Quantification of the expression of gcy-5$^{prom}$::gfp in various tissues after CHE-1$^{hs}$ induction at various ages, as listed on the X-axis. n => 20 for each stage. (C) Expression of gcy-6$^{fosmid}$ and gcy-7$^{prom}$ in response to CHE-1$^{hs}$ at different stages. Quantification of the ectopic expression: every dot in this graph represents a worm and the number on the y-axis represents the total number of marker-expressing neurons counted. Ectopic expression decreases as the age of the embryos being heatshocked increases. (D) A synthetic ~300 bp region of the gcy-5 promoter in which the ASE motif has been multimerized four times can be broadly induced by CHE-1$^{hs}$ at all stages, including L4s, as shown here. Some of the tissues that express gcy-5$^{4XASE}$::gfp are labeled. gcy-5$^{4XASE}$::gfp is normally expressed only in the two ASEs.

The following figure supplements are available for figure 1:

**Figure supplement 1.** Tools used for the detection of gcy-5 expression.

**Figure supplement 2.** Expression of gcy-5 in response to CHE-1$^{hs}$ at different developmental stages.

**Figure supplement 3.** Expression of ASE markers in response to cell type-specific, but not temporally controlled, che-1 expression.

normal development (*Yu et al., 1997*). If CHE-1$^{hs}$ is induced during early embryonic stages, the resulting arrested embryos and larvae activate broad expression of gcy-6$^{fosmid}$::gfp and gcy-7$^{prom}$:: rfp. Expression of these markers also becomes more restricted as CHE-1$^{hs}$ is induced at later stages. However, ectopic induction of these genes is not as broad as gcy-5 and no ectopic expression is detected when CHE-1$^{hs}$ is induced in larval stages. (*Figure 1C*). Previous analysis has shown that while CHE-1 is sufficient to activate ASER-specific genes, it may require other co-activators to activate ASEL-specific genes (*Etchberger et al., 2009*; *Hobert, 2014*). This may explain the difference in the breadth of ectopic CHE-1 activity.

Since embryonic expression of CHE$^{hs}$ leads to deformation and developmental arrest, we expressed CHE-1 using tissue-specific promoters to understand more thoroughly its activity in various differentiating cell types. We utilized a muscle (*unc-27$^{prom}$*), hypodermal (*dpy-7$^{prom}$*), pan-neuronal (*ric-19$^{prom}$*), and two neuron-type-specific drivers (*unc-47$^{prom}$* expressed in GABAergic MNs, *ift-20$^{prom}$* expressed in sensory neurons). With the exception of the hypodermal promoter, these promoters all become activated only in post-mitotic cells at the time that cells are initiating expression of various identity-specific genes. CHE-1 driven by all neuronal promoters resulted in broad ectopic expression of gcy-5$^{prom}$::gfp, gcy-5$^{fosmid}$::gfp, and ceh-36$^{fosmid}$::yfp (another che-1 target, a homeodomain TF expressed in ASEL/R and AWC neurons) . However, gcy-6$^{fosmid}$::gfp and gcy-7$^{prom}$::rfp show restricted activation in only 1–2 extra cells (*Figure 1—figure supplement 3A*). Transgenic animals expressing CHE-1 in the hypodermis or muscle also showed gcy-5$^{prom}$::gfp expression in the respective tissues (*Figure 1—figure supplement 3b,c*). In combination with the heatshock experiments, these experiments show that a wide range of cell types are receptive to CHE-1 activity after post-mitotic division while they are terminally differentiating. Once terminal differentiation is accomplished, most cells lose their multipotency and become refractory to CHE-1 activity.

## Terminal selector *unc-3* restricts plasticity of cholinergic motor neurons

Our finding that cells lose their receptivity to ectopically expressed CHE-1 after terminal differentiation led us to hypothesize that the loss of cellular plasticity might be coordinated with the acquisition of a terminal identity. In *C. elegans*, the expression of identity-specific genes in many neurons is established by single or combinations of terminal selector TFs. These TFs are generally expressed after the terminal cell division and are required for both the activation and maintenance of identity-specific effector gene batteries (*Hobert, 2011*). In the absence of such TFs, the mutant neurons are born and express numerous pan-neuronal genes that normally get activated at the same time as identity-specific genes, suggesting that multiple aspects of terminal differentiation proceed normally. However, the lack of identity- and function-specific genes which include neurotransmitter synthesis genes, ion channels, etc., indicate that these neurons fail to acquire their neuron-type-specific identity. We asked whether, in addition to specifying identity, these terminal selector TFs might also facilitate the process of restriction, thereby coordinating it with terminal differentiation. If this were

the case, in the absence of a terminal selector gene, a cell would remain in a more plastic state, and in our experimental paradigm, retain its receptivity to CHE-1[hs]. Alternatively, if an independent mechanism triggers the restriction of cellular plasticity, cells would lose their receptivity to CHE-1[hs] regardless of the presence of the terminal selector.

To test these alternative hypotheses, we induced CHE-1[hs] in *unc-3* mutants. *unc-3* is a COE-type TF that acts as the terminal selector for A/B-type and AS cholinergic MN identity (*Kratsios et al., 2011*; *Prasad et al., 1998*). *unc-3(-)* cholinergic MNs are born and found in approximately their wild-type positions in the ventral nerve cord (VNC), retro-vesicular ganglion (RVG), and pre-anal ganglion (PAG) of the worm. They express pan-neuronal identity markers, and form axonal projections. However, these neurons lack expression of all the acetylcholine pathway genes, numerous other terminal identity genes, show defects in synaptogenesis and are non-functional (*Kratsios et al., 2011*; *Prasad et al., 1998*). To test if such *unc-3(-)* neurons retain plasticity, we induced CHE-1[hs] in wildtype and *unc-3* mutant worms at various larval stages and scored the expression of *gcy-5[prom]::gfp* in the RVG, VNC, and PAG. In addition to fifty *unc-3* dependent MNs, these ganglia contain 19 GABAergic MNs and 13 other *unc-3*-independent neurons (*Figure 2A*). To decrease the ambiguity in scoring, we used a red fluorescent GABAergic marker in the background and counted the number of *gcy-5*-positive cells that were non-GABAergic. In wildtype worms, we find that an average of 11 neurons activate *gcy-5[prom]::gfp* at the L2 stage upon ectopic CHE-1[hs] expression, and this number is reduced to an average of 2 neurons by the L4 and adult stages. In *unc-3* mutants, a significantly higher number of neurons express *gcy-5[prom]::gfp* at every developmental stage upon ectopic CHE-1[hs] expression (*Figure 2B*). *gcy-5[prom]::gfp* expression in the VNC was not seen when *unc-3* mutants were heatshocked in the absence of the *hs[prom]::che-1* transgene, showing that the loss of *unc-3* is not sufficient for an identity switch. We repeated these experiments using the single copy *heatshock[prom]::che-1* transgene, and again observed that *gcy-5[prom]::gfp* expression is induced in significantly more *unc-3(-)* than wildtype cells. Additionally, smFISH of these worms showed that expression of endogenous *gcy-5* mRNA was also induced in *unc-3(-)* cells to a similar extent (*Figure 2C*). On average, about 30% of affected neurons in *unc-3* mutants express *gcy-5* in response to CHE-1[hs] induction.

To assess the breadth of CHE-1[hs] activity, we also examined the activation of other CHE-1 target genes in the VNCs of *unc-3* mutants. Genes that showed increased induction after L4 CHE-1[hs] in *unc-3* included the glutamate transporter *eat-4[fosmid]::yfp*, the homeodomain TF *ceh-36[fosmid]::yfp*, and the ciliary protein *ift-20[prom]::rfp* (*Figure 2D*). No induction of *gcy-6[fosmid]::gfp* or *gcy-7[prom]::rfp* was observed in wildtype or *unc-3(-)* neurons, which is consistent with the idea that *che-1* is not sufficient for the induction of these genes, as discussed earlier (0 neurons induced in >20 worms checked). As the transgenic markers that show increased induction in *unc-3(-)* neurons are randomly integrated into the different chromosomes of the worm, this phenotype suggests that several regions of the genome remain receptive to CHE-1[hs] activity in the mutant condition.

*unc-3* is required broadly for activation of both generic and sub-type-specific cholinergic MN identity features. The further diversification of cholinergic MNs into sub-types is controlled by the activity of various class-specific repressors, such as paired homeodomain TF *unc-4*, the nuclear receptor TF *unc-55*, and the T-box TF *mab-9* (*Kerk et al., 2017*; *Miller and Niemeyer, 1995*; *Pocock et al., 2008*). The loss of these TFs results in MNs with mixed class identities and subsequent functional defects. We asked if these repressor TFs also contribute to the restriction of plasticity. Neither *unc-4* mutants, nor *unc-55; mab-9* double mutants, however, show increased induction of *gcy-5[prom]::gfp* in cholinergic MNs in response to CHE-1[hs] (*Figure 3—figure supplement 1A*). These data demonstrate that a disruption of function and identity-specific transcriptional programs does not automatically make cells more plastic.

## Restriction of cellular plasticity is a function shared by several terminal selectors

We examined the expression of *gcy-5* in response to adult CHE-1[hs] after removal of six other terminal selectors that control the identity of a wide range of different neuron types in the head and tail ganglia of the worm. Four of the six mutants showed a phenotype similar to *unc-3* in that their removal allowed ectopic, CHE-1[hs]-dependent *gcy-5* expression. These terminal selectors are ETS-domain TF *ets-5,* and homeodomain TFs *ttx-1*, *ceh-14*, and *lin-11*. Specifically, *ets-5* is a terminal selector for the $CO_2$ sensing BAG neurons (*Guillermin et al., 2011*). In our assay, 30% of *ets-5(-)*

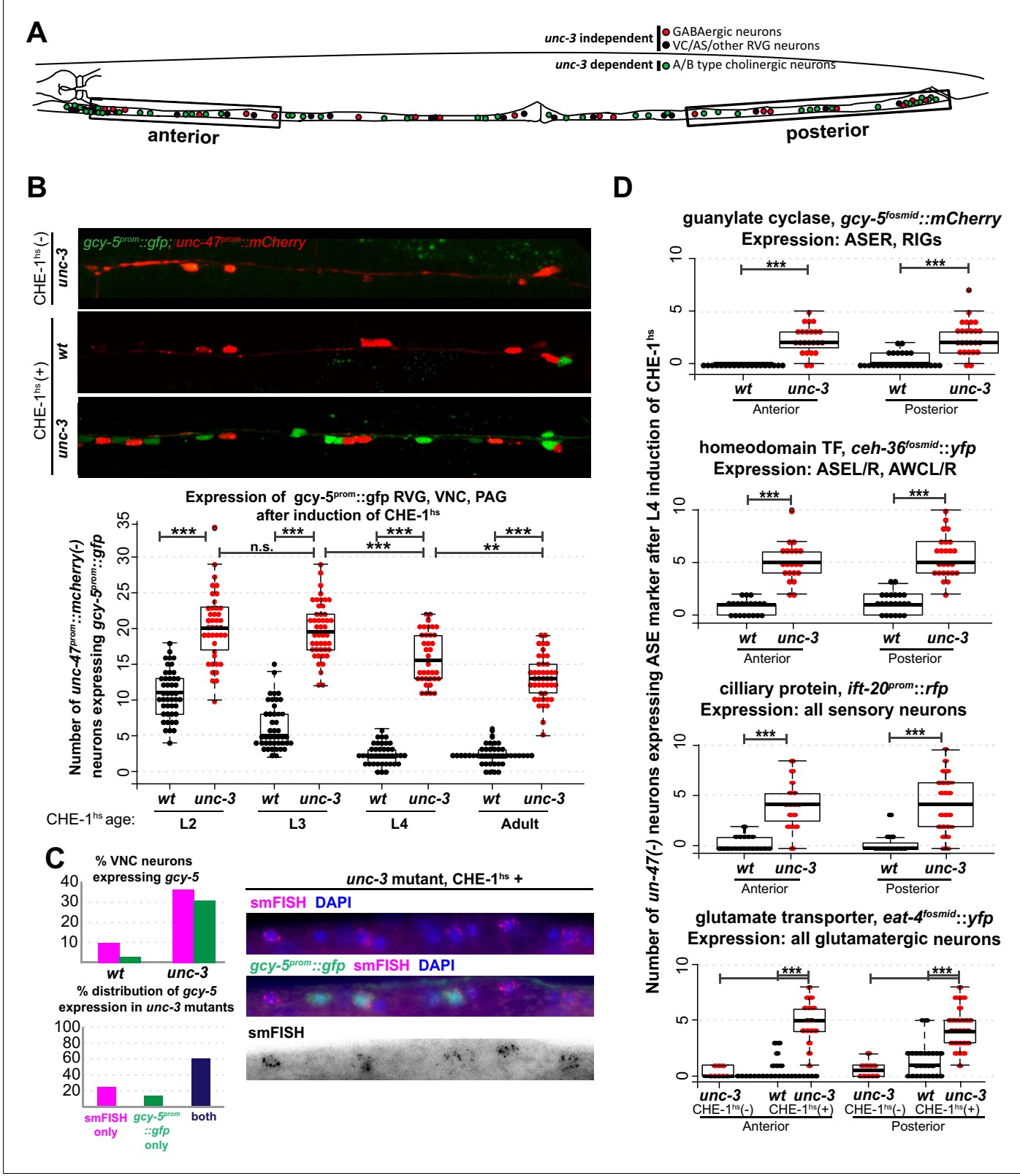

**Figure 2.** Several CHE-1 targets retain their CHE-1 responsiveness in *unc-3* mutant cholinergic MNs. For all heatshock experiments, three biological replicates were performed. The student t-test is used for statistical comparisons, *p<0.05, **p<0.01, ***p<0.001, n.s.= p>0.05. (**A**) A schematic representation of the neurons in the RVG, VNC, and PAG of the worm. In heatshock experiments represented in B, C, and D, the number of ASE marker expressing cells in the RVG, VNC, and PAG were scored. In B and D, *unc-47*, a marker of GABAergic identity (is used in the background and

*Figure 2 continued*

only *unc-47(-)* neurons are scored. This strategy ensures proper scoring of the ~50 *unc-3* dependent cholinergic MNs . (B) Images of wildtype and *unc-3* worms with and without CHE-1$^{hs}$ and a quantification of *gcy-5$^{prom}$* induction at various stages of development. Every dot in this plot represents an individual worm. The entire RVG, VNC, and PAG of the worm is scored. A significantly higher number of *gcy-5$^{prom}$::gfp* neurons are seen in *unc-3* mutants as compared to wildtype in response to CHE-1$^{hs}$ induction. No *gcy-5$^{prom}$::gfp* neurons are seen in *unc-3* mutant worms that get heatshocked but do not contain the *heatshock$^{prom}$::che-1* array (0 cells in >20 worms). (C) *gcy-5$^{prom}$* transgene (green bar) and *gcy-5* mRNA (smFISH; pink bar) induction in heatshocked worms carrying a single copy insertion of *heatshock$^{prom}$::che-1*. Induction of *gcy-5* endogenous mRNA (smFISH) is similar to the induction of the *gcy-5$^{prom}$* reporter. (D) Expression of various other ASE markers is seen in a significantly higher number of *unc-47(-)* RVG, VNC, and PAG neurons in *unc-3* mutants as compared to wildtype after CHE-1$^{hs}$ induction. The scored markers and their wildtype (without CHE-1$^{hs}$) expression patterns are listed above each plot. Every dot in these plots represents an individual worm. Markers are scored only in the anterior and posterior regions of the worm, as represented in A. These worms were heatshocked at the L4 stage.

BAG neurons express *gcy-5$^{fosmid}$::gfp* in response to adult CHE-1$^{hs}$ as opposed to 4% of wildtype BAG neurons (*Figure 3A*). *ttx-1* is a terminal selector for thermosensory AFD neurons (*Satterlee et al., 2001*; *Serrano-Saiz et al., 2013*) and 23% of *ttx-1(-)* AFD neurons express *gcy-5$^{fosmid}$::gfp* after adult CHE-1$^{hs}$ whereas 2% of wildtype AFD neurons do (*Figure 3B*). Removing both *ttx-1* and *ceh-14*, the co-factor of *ttx-1* in AFD specification, results in a 34.3% phenotype compared to 4.8% in matched wildtypes. *ceh-14* also acts as a terminal selector in several distinct tail neuron classes (*Serrano-Saiz et al., 2013*). We found an average increase of five *gcy-5$^{fosmid}$*-positive neurons in the tails of *ceh-14* mutants upon CHE-1 induction (*Figure 3C*). Lastly, *lin-11* is required for terminal identity specification of ASG neurons (*Serrano-Saiz et al., 2013*), and 30% of *lin-11(-)* ASG neurons express *gcy-5$^{fosmid}$::gfp* as opposed to 2% of wildtype ASG neurons (*Figure 3d*).

Two terminal selector mutants, the homeodomain TFs *unc-30* and *unc-42,* did not share this phenotype. *unc-30* is the terminal selector for GABAergic MNs in the VNC and *unc-42* is the terminal selector for ASH sensory neurons (*Baran et al., 1999*; *Cinar et al., 2005*; *Jin et al., 1994*; *Serrano-Saiz et al., 2013*). No increases in induction of *gcy-5$^{prom}$::gfp* or *ceh-36$^{fosmid}$::yfp* were detected in response to CHE-1$^{hs}$ in the GABAergic MNs of *unc-30* (*Figure 3—figure supplement 1C*, data only shown for *ceh-36$^{fosmid}$::yfp*). In the case of *unc-42* mutants, neither *gcy-5$^{fosmid}$::rfp*, nor *ceh-36$^{fosmid}$::yfp* was detected in the ASH neurons after CHE-1 misexpression (*Figure 3—figure supplement 1D*, data only shown for *gcy-5$^{fosmid}$::rfp*). Expression of *che-1* early in the development of GABAergic MNs and ASH neurons does lead to activation of target genes, based on tissue-specific promoter driven expression of *che-1* (*Figure 3—figure supplement 1A*). Additionally, activation of CHE-1$^{hs}$ in L4s/adults results in the expression of *gcy-5$^{4XASE}$::gfp* in both sets of neurons (*Figure 3—figure supplement 1E,F*). This means that CHE-1 is competent to activate target genes in these neurons while they are differentiating, and that its activity as a TF is uncompromised in these neurons during the adult stage. Therefore, we conclude that while *unc-30* and *unc-42* are required for terminal differentiation, they are dispensable for restricting cellular plasticity.

Overall, these results suggest that numerous terminal selector genes play a role in restricting the activity of ectopically expressed CHE-1. The affected neurons in all these TF mutants lack expression of certain identity-specific effector genes. The negative results with *unc-30* and *unc-42* demonstrate that the loss of a terminal identity in itself it not sufficient to keep a neuron in a plastic state. This suggests that the restriction of plasticity is a distinct and parallel function carried out by certain terminal selector TFs. In the next sections, we provide more evidence that supports this functional distinction.

## The identity-specifying and plasticity-restricting functions of UNC-3 can be genetically separated

Two alternative hypotheses can explain the function of terminal selectors in the restriction of plasticity. Terminal selectors could simply be required for terminal differentiation, and it would be the completion of differentiation that then indirectly triggers restriction of plasticity. Alternatively, these transcription factors could be required for restricting plasticity in a mechanism that is separate from inducing effector genes of terminal cellular identity. A hypomorphic allele of *unc-3* that we inadvertently generated by CRISPR-Cas9 mediated transgenesis allowed us to separate the identity-specifying function from the plasticity-restricting function of *unc-3*. This allele contains an insertion of

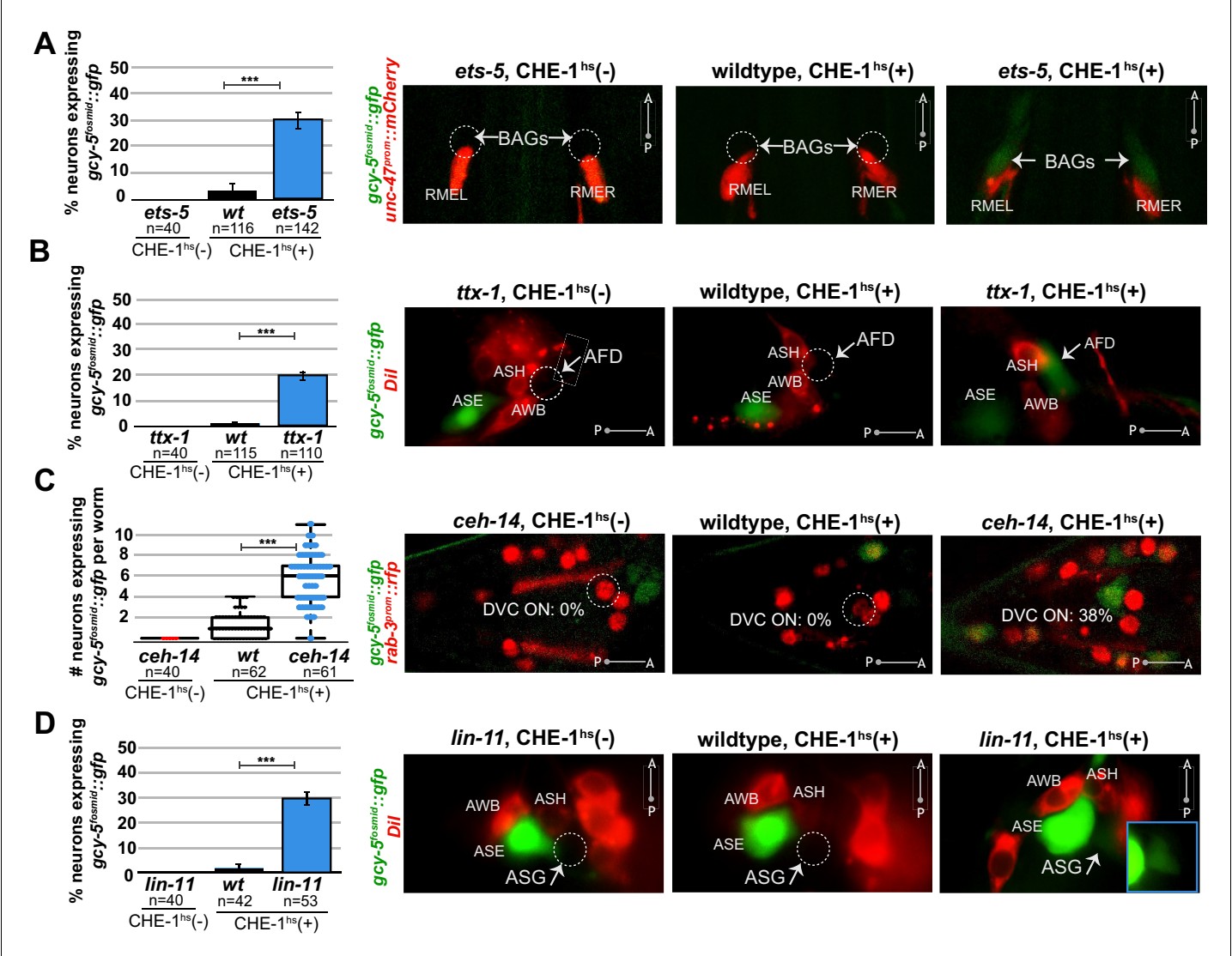

**Figure 3.** Several distinct terminal selectors control restriction of cellular plasticity in distinct neuron types. CHE-1[hs] was induced in adults for all experiments except *lin-11*, for which L4s were heatshocked (*lin-11* mutant adults are very short-lived due to internal hatching of progeny). The markers in red are used to help identify the neuron of interest. *gcy-5[fosmid]::gfp* is more frequenctly activated by CHE-1[hs] in BAG neurons of *ets-5* mutants (panel A), in AFD neurons of *ttx-1* mutants (panel B), in DVC and a few other unidentifiable neurons in the tail of *ceh-14* mutants (panel C), and in ASG neurons of *lin-11* mutants (panel D). All error bars represent SEM. Data are accumulated from three independent heatshock experiments. The student t-test is used, ***p<0.0001, n.s.= p>0.01.

The following figure supplement is available for figure 3:

**Figure supplement 1.** Phenotypes of additional mutants.

a plant-specific AID degron into the *unc-3* locus that allows for protein degradation in the presence of exogenous plant hormone auxin and the Arabidopsis F-box protein *TIR1* (*Zhang et al., 2015*) (*Figure 4A*). In the absence of TIR1 and auxin, the *unc-3::mNG::AID* worms display wildtype locomotion and express cholinergic MN markers at near wildtype levels (*Figure 4A,B*). However, cholinergic MNs of these worms were receptive to CHE-1[hs], even in the absence of TIR1 and auxin (*Figure 4C,D*). When CHE-1[hs] is overexpressed in adult worms containing *unc-3::mNG::AID*, expression of *gcy-5[prom]::gfp* is observed at a penetrance indistinguishable from that observed in the *unc-3* null allele. Hence, in this allele the cholinergic MNs differentiate largely normally, and yet retain

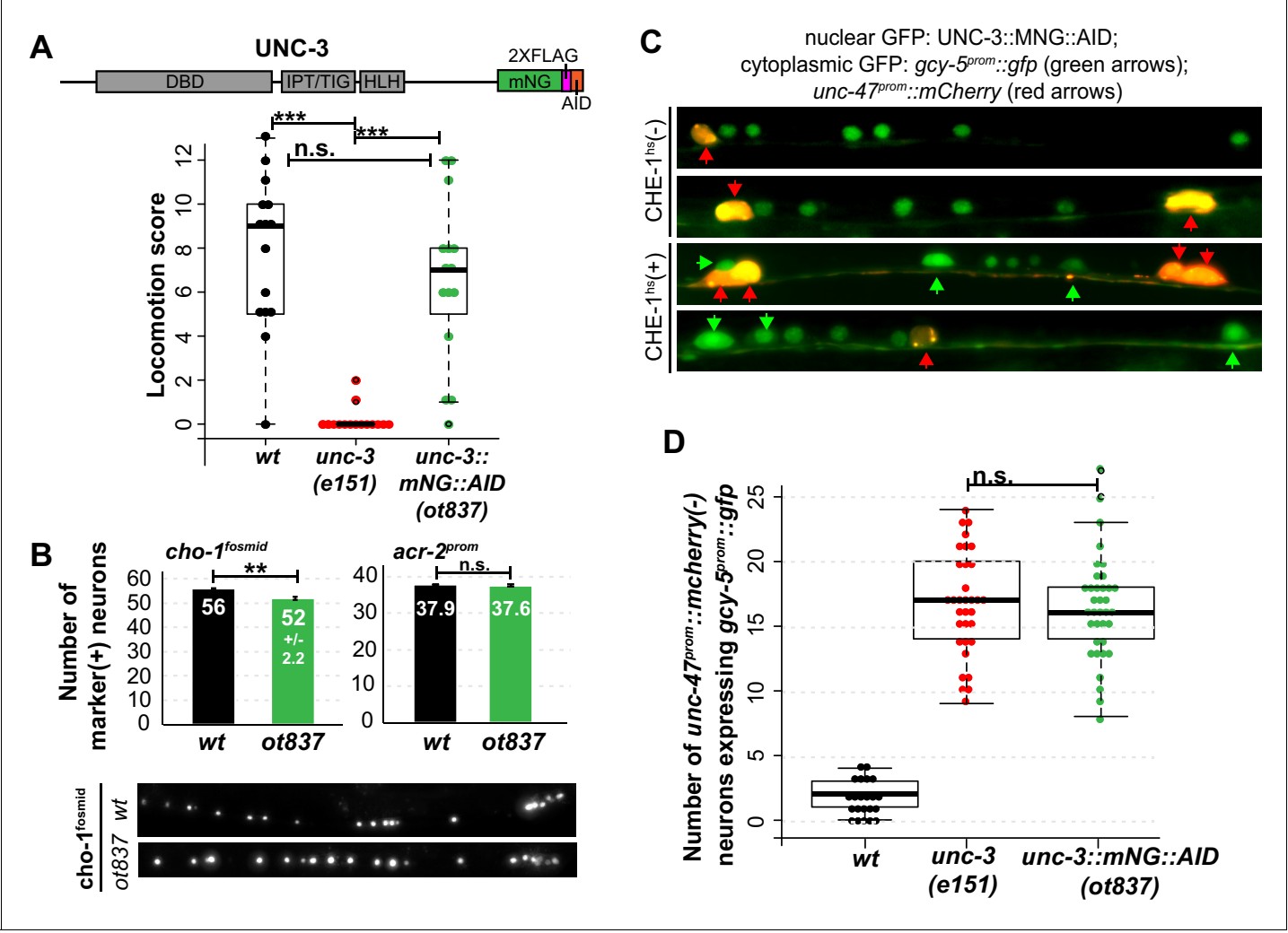

**Figure 4.** Genetic separation of identity-specification and plasticity restriction. All error bars represent SEM. The student t-test is used, *p<0.05, **p<0.01, ***p<0.001, n.s.= p>0.05. (A) An mNeonGreen::AID tagged allele of *unc-3*, *ot837*, does not have the locomotion defects of an *unc-3* null allele. (B) *ot837* worms show near wildtype and wildtype induction of *cho-1fosmid* and *acr-2prom* respectively, whereas in *unc-3* null mutants these genes are strongly repressed (**Kratsios et al., 2011**). *cho-1fosmid* expression is missing in ~4 neurons in *ot837*, but levels of expression in the neurons that retain expression look wildtype. (C) *ot837* behaves like the *unc-3* null with respect to restriction of plasticity. Induction of *gcy-5prom* is indistinguishable in *ot837* and the *unc-3* null allele after CHE-1hs induction. (D) Images of *gcy-5prom* expression in *ot837* worms after CHE-1hs induction. In these images the nuclear green signal comes from the mNG tagged UNC-3 protein, the cytoplasmic GFP from *gcy-5prom* (green arrows), and the red cells are GABAergic (red arrows).

plasticity in response to CHE-1hs activity. This allele therefore supports the hypothesis that terminal selector-like TFs control restriction independently of specification. While it is unclear how the AID allele affects the *unc-3* protein, we propose that this allele results in lower levels of active protein, perhaps by degradation through an auxin/TIR1 independent pathway.

### *unc-3* regulates chromatin compaction

We next asked if terminal selectors restrict cellular plasticity by controlling chromatin accessibility at genes that are not expressed in differentiated cells. We focused again on *unc-3* and hypothesized that in the wildtype cholinergic MNs CHE-1 targets would be present in a repressed, inaccessible chromatin state while in *unc-3* mutants they would be present in a more open state, such that they are more likely to be activated by CHE-1. To visualize the chromatin state with single cell resolution, we used a chromosome tagging method that has previously been used to study the localization and

state of compaction of transgenic loci in *C. elegans* (*Cochella and Hobert, 2012*; *Fakhouri et al., 2010*; *Meister et al., 2010*; *Yuzyuk et al., 2009*). We created a repetitive transgene array containing *gcy-5^prom^::rfp* and bacterial LacO sites and randomly integrated it into the genome. These transgenic worms also carry a separate transgene to ubiquitously express a LacI::GFP fusion protein. Thus, in every nucleus of this worm, LacI::GFP binds to the LacO sites surrounding the *gcy-5* reporter, thereby labeling the *gcy-5* transgenic loci as two green dots (*Figure 5A,B*). We also generated LacO arrays for an *unc-3*-dependent cholinergic gene, *ace-2^prom^::rfp,* as a control (*Figure 5C*). This technique has previously been used to show a correlation between the compaction state of the LacI::GFP spots and the repressed or active state of the locus in the array (*Cochella and Hobert, 2012*; *Fakhouri et al., 2010*; *Meister et al., 2010*; *Yuzyuk et al., 2009*).

To facilitate quantification of the state of compaction, wildtype and *unc-3* mutant animals containing the *ace-2* and *gcy-5* LacO arrays were fixed with paraformaldehyde and stained with GFP and DAPI. The total number of GFP pixels per VNC neuron was then calculated as an approximation of the space occupied by the *ace-2* and *gcy-5* LacO transgenes in wildtype and *unc-3* mutant neurons. This quantification was performed blinded. The *ace-2* LacO array is transcriptionally active in wildtype cholinergic neurons but not in *unc-3* mutants (*Figure 5B,C*). Consistent with its expression, the *ace-2* LacO array showed a significantly higher GFP pixel count in wildtype VNC neurons as compared to *unc-3* (*Figure 5C,D*). By showing that the *ace-2* transgenic locus is more open when it is transcriptionally active, this experiment indicated that the LacI/O dot assay is indeed indicative of the state of chromatin in cholinergic MNs. We then quantified GFP spots on the LacO array that contains *gcy-5*, the ASE-specific CHE-1 target gene that is normally not expressed in VNC MNs. In this case, the size of the GFP spot was increased in *unc-3* mutant MNs compared to wildtype (*Figure 5E*). These data show that the *gcy-5* locus is in a less compact state in *unc-3* mutant cholinergic MNs, which may make it more amenable to activation by CHE-1. This change in compaction was not as dramatic as the one observed in the *ace-2* array, but this may be because while the *ace-2* array is actively transcribed in wildtype neurons, the *gcy-5* array is not being actively transcribed in *unc-3* mutants, rather it is simply in a more accessible state so that it is more likely to be transcribed when CHE-1 is ectopically induced. The quantified worms were also co-stained with H3K27me3 to control for staining efficiency and we did not observe a significant change in the number of H3K27me3 pixels between wildtype and *unc-3* in the same neurons (*Figure 5F*).

If the change in compaction of the *gcy-5* array were caused by *unc-3* function, this change should be specific to neurons in which *unc-3* acts as a terminal selector. Indeed, the size of the *gcy-5* LacO array in hypodermal cells of wildtype and *unc-3* mutants show no statistically significant difference (*Figure 5F*). It is also important to note that not all VNC neurons contain larger GFP spots, which is consistent with the variability we observe in the number of cells per worm that activate *gcy-5* in response to CHE-1 (*Figure 5E*, *Figure 2B*).

## Interplay between *unc-3* and chromatin modifiers in regulating the restriction of plasticity

To confirm and further understand the role of chromatin compaction in our restriction assay, we probed the function of H3K27me and H3K9me. Both of these modifications are associated with silent and condensed chromatin (*Towbin et al., 2012*; *Yuzyuk et al., 2009*).

We have previously shown that the removal of the *C. elegans* PRC2 complex, which functions as an H3K27methyl transferase, makes germ cells more plastic and receptive to CHE-1 activity (*Patel et al., 2012*). Here, we tested the activity of CHE-1 in the ventral nerve cord of *mes-2*/E(z) mutants. *mes-2* is the enzyme responsible for the histone methyl transferase activity of PRC2 and is known to be required for H3K27me3 in the *C. elegans* germline at all stages, and in somatic cells during embryonic and early larval stages, but not in the adult (*Bender et al., 2004*). We found that loss of *mes-2* permits CHE-1^hs to induce *gcy-5^prom^::gfp* to the same extent as removal of *unc-3* does (*Figure 6A*), indicating that *mes-2* is indeed required for restriction of plasticity in cholinergic MNs. To determine whether *mes-2* acts in parallel or in the same pathway as *unc-3*, we tested *mes-2; unc-3* double mutants and found that they have an additive phenotype. Removal of both *unc-3* and *mes-2* leads to the induction of *gcy-5^prom^::gfp* in twice as many cells as seen in *unc-3* or *mes-2* mutants alone. This suggests that *unc-3* and *mes-2* work in parallel genetic pathways to ensure plasticity restriction in differentiated cholinergic MNs.

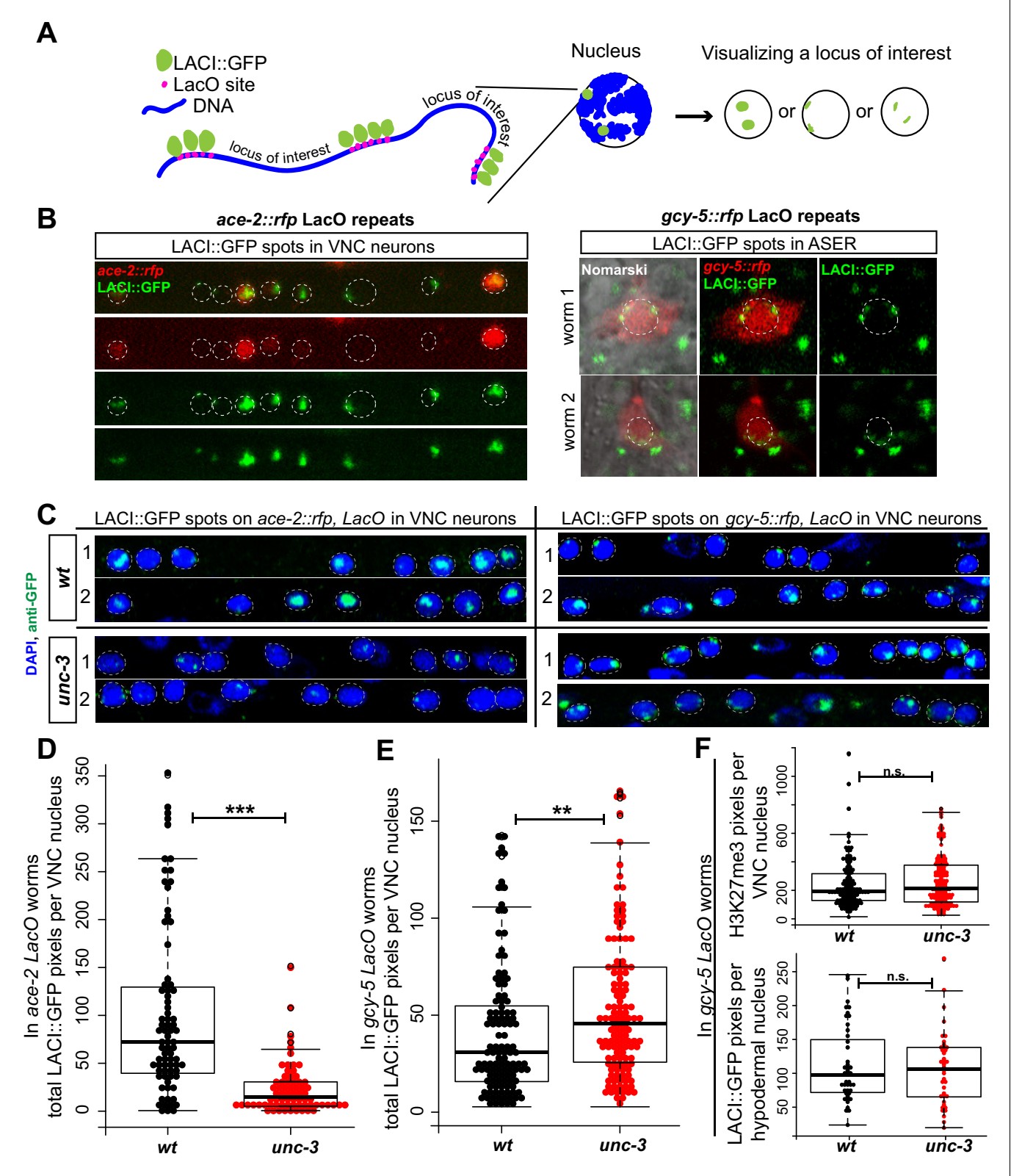

**Figure 5.** Assessing chromatin state of a *gcy-5* transgenic locus with the LacI/LacO dot assay. The student t-test is used for statistical comparisons, *p<0.05, **p<0.01, ***p<0.001, n.s.= p>0.05. (A) Schematic representation of the spot assay. A transgene containing the *gcy-5^{prom}::rfp* sequence and LacO binding sites is integrated into the genome of worms that ubiquitously express LacI::GFP. In the resulting transgenic worm, every nucleus contains two GFP spots representing the LacI::GFP bound to the *gcy-5* LacO array. The structure and location of this locus can be visualized by GFP. (B) The
*Figure 5 continued on next page*

Figure 5 continued

*gcy-5* arrays localize to the nuclear periphery in all cells in which expression was checked, including the ASEs. The array is transcriptionally active in the ASER, as evident by the presence of RFP, suggesting that this locus localizes to the nuclear periphery regardless of its transcriptional state. An identically built *ace-2*[prom]*::rfp* LacO array shows sub-nuclear localization that correlates to its transcriptional state. In cells that express *ace-2*[prom]*::rfp*, as seen by the presence of RFP, the GFP spots are diffused and in the nuclear lumen, suggesting that the locus is decompacted. In cells that do not express *ace-2*[prom]*::rfp*, the GFP spots are compact and localized to the nuclear periphery. Two independent array integrants examined for both *ace-2* and *gcy-5* showed similar results and one is shown here. (C) Worms were fixed and stained with DAPI for quantification of the GFP spots as an estimation of array structure. (D) *ace-2* arrays were significantly larger when transcriptionally active as opposed to inactive in the VNC neurons of wildtype worms compared to *unc-3* mutants. In (D, E, F) each dot represents a single nucleus and the number on the y-axis is the total pixel count occupied by the LacI/O spots or H3K27me3 per neuron. (E) *gcy-5* arrays were significantly larger in *unc-3* mutant neurons than in wildtype. (F) Control staining for H3K27me3 in *gcy-5* LacO array containing worms showed no difference between wildtype and *unc-3* and the size of the *gcy-5* LacI/O spots do not change in the hypodermal nuclei of wildtype vs. *unc-3* worms.

To test the role of H3K9me3, we tested the ability of CHE-1[hs] to activate *gcy-5* in the VNC of cholinergic MNs in *met-2* and *set-25* mutants. *met-2* and *set-25* are SET-domain proteins shown to be required for H3K9 methylation in *C. elegans* embryos, young larvae, and adult germ lines (*Towbin et al., 2012*; *Zeller et al., 2016*). *met-2* mutant embryos lack H3K9me1/2/3 whereas *set-25* embryos lack H3K9me3 (*Towbin et al., 2012*). Using antibody staining, we ensured that *met-2; set-25* double mutants also lack H3K9me3 at the L4 and adult stages where we perform our experiments (*Figure 6—figure supplement 1A*). If H3K9me3 plays a role in decreasing the accessibility of a *gcy-5*[prom]*::gfp*, then this locus should remain more accessible in *met-2* and *set-25* mutants. We found that induction of CHE-1[hs] does lead to increased activation of *gcy-5*[prom]*::gfp* in the cholinergic MNs of *met-2* single mutants and *met-2 set-25* double mutants, but not in *set-25* single mutants (*Figure 6B*). This suggests that H3K9me1/2 and H3K9me3 have to be perturbed for CHE-1[hs] to activate *gcy-5*[prom]*::gfp*. H3K9me3 is known to be involved in the repression of repetitive loci. To ensure that our phenotype is not an artifact of repetitive transgenes, we confirmed that the loss of *met-2; set-25* in transgenic worms containing the single copy *hs::che-1* transgene also results in the expression of *gcy-5*[prom]*::gfp* in the cholinergic MNs and that the expression of *gfp* in these worms overlaps with *gcy-5* mRNA expression as determined by smFISH of endogenous *gcy-5* transcripts (*Figure 6E*).

To test genetic interactions between *unc-3*, *met-2* and *set-25* we examined CHE-1[hs] dependent induction of *gcy-5*[prom]*::gfp* in *met-2;unc-3*, *set-25;unc-3* and *met-2 set-25;unc-3* double and triple mutants. None of these double and triple mutants show an additive effect that is significantly different from the phenotype of *unc-3* single mutants (*Figure 6B*), indicating that *met-2, set-25,* and *unc-3* act in the same pathway rather than in parallel pathways. Since the removal of *met-2* and *set-25* does not result in as strong a phenotype as an *unc-3* single mutant, we furthermore infer that *met-2; set-25* dependent H3K9 methylation is not the only pathway affected in the *unc-3* mutants.

To investigate the mechanisms of H3K9 action further, we considered HP1, a conserved chromodomain containing protein that is known to bind methylated H3K9 (*Bannister et al., 2001*; *Jacobs and Khorasanizadeh, 2002*; *Jacobs et al., 2001*; *Lachner et al., 2001*; *Nielsen et al., 2002*). The *C. elegans* genome encodes two homologs of HP1, *hpl-1* and *hpl-2* (*Couteau et al., 2002*; *Schott et al., 2006*). *hpl-1* has been shown to colocalize with H3K9me3 in embryos by immunofluorescence, and *hpl-2* shows enriched binding at H3K9me1/2 regions by whole embryo ChIP analysis (*Garrigues et al., 2015*; *Vandamme et al., 2015*). Fluorescent-tagged HPL-1 and HPL-2 proteins localize to distinct foci on chromatin and these proteins seem to have both unique and redundant functions (*Schott et al., 2006*). We asked if mutants of these proteins phenocopy the loss of *met-2* and *unc-3* in MNs after CHE-1[hs] expression. Indeed, an increased induction of *gcy-5*[prom]*::gfp* was seen in MNs after removal of both *hpl-1* and *hpl-2*, and this phenotype was enhanced in the double mutant, so that it was stronger than the phenotype of *met-2 set-25* mutants (*Figure 6c*). An *hpl-2; hpl-1 unc-3* triple mutant also shows a phenotype that is enhanced from the *hpl-1; hpl-2* double or the *unc-3* mutant alone, but it is not completely additive. This enhancement indicates that HP1 has functions independent of *met-2 set-25* and H3K9me. Induction of *gcy-5* in all of the mutants tested here required CHE-1[hs] induction. Taken together, both H3K9me and H3K27me contribute to the restriction of plasticity in differentiated cells. H3K9me might be acting along with terminal selectors to mediate this process, while H3K27me acts in parallel (*Figure 7*).

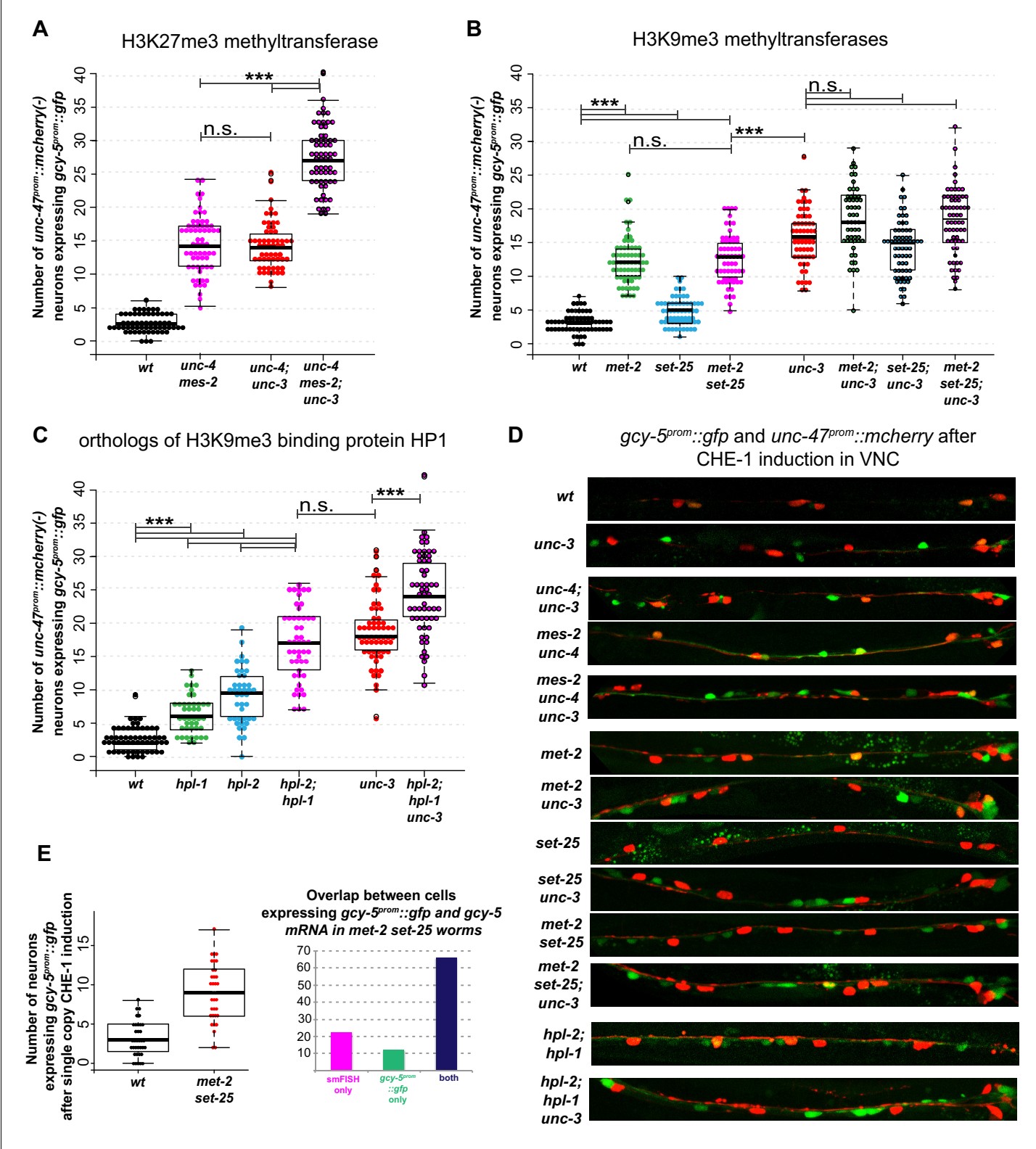

**Figure 6.** Interactions of *unc-3* with chromatin mutants. CHE-1[hs] was induced at the L4 stage. For each graph, the data cumulatively represent three independent heatshock experiments in which all genotypes were scored in parallel. Every dot represents an individual worm. One way ANOVA with post-hoc Bonferroni and Scheffé tests for multiple comparisons were performed independently for each group of strains tested together. *p<0.05, **p<0.01, ***p<0.001, n.s.= p>0.05. (**A**) *mes-2* mutants show increased induction of *gcy-5[prom]::gfp* in response to CHE-1[hs] and show an additive

*Figure 6 continued on next page*

*Figure 6 continued*

phenotype with *unc-3*, suggesting H3K27me3 regulates restriction in a parallel pathway with *unc-3*. The *mes-2* and *unc-4* mutations are genetically linked. *unc-4* does not play a role in restriction, but it was used in all strains tested in this set of experiments to ensure similar genetic backgrounds. (B) Both *met-2* and *set-25* mutants have significantly more *gcy-5^prom^::gfp* expressing cholinergic MNs compared to wildtype. However, for *set-25* the increase is only modest, and a *met-2 set-25* double mutant is not significantly different from *met-2*, suggesting *met-2*, has a greater role in making the *gcy-5^prom^::gfp* locus inaccessible to CHE-1. An additive phenotype is not evident in *met-2; unc-3, set-25; unc-3*, or *met-2 set-25; unc-3*, suggesting that these factors might act in the same pathway to restrict the accessibility of *gcy-5^prom^::gfp*. No ectopic expression of *gcy-5^prom^::gfp* is seen in these mutants in the absence of CHE-1^hs^ (not shown, n > 50 worms). (C) HP1 mutants also show increased induction of *gcy-5^prom^::gfp*, and slightly enhance the phenotype of *unc-3* suggesting they have roles independent of *met-2 set-25* dependent H3K9me. No *gcy-5^prom^::gfp* is induced in *hpl-1;hpl-2* double mutants in the absence of CHE-1^hs^ (not shown, n > 50 worms). (D) Representative images of scored worms. (E) Induction of CHE-1^hs^ with the single copy transgene also results in increased induction of *gcy-5^prom^::gfp* in the VNC of *met-2 set-25* mutants and the number of *gcy-5^prom^::gfp* and *gcy-5* mRNA expression cells in the double mutant worms are similar showing that the phenotype in (B) is not simply a result of derepressed repetitive transgenes.

The following figure supplement is available for figure 6:

**Figure supplement 1.** H3K9 in wildtype and *unc-3* mutant worms.

We hypothesized that perhaps in *unc-3* mutants, a reduction of histone modifications that are associated with active chromatin marks at *unc-3*-induced terminal differentiation genes might be causing a titration of negative histone marks over the genome, thereby increasing accessibility at genes that would otherwise have high levels of negative marks like H3K9me3. It has, for example, been observed that removal of positive mark H3K36 trimethylation leads to a titration of H3K27me3 from loci that are normally repressed (*Gaydos et al., 2012*). However, genetic removal of positive histone marks like H3K4me3 and H3K3me3 did not result in enhanced cellular plasticity (*Figure 6— figure supplement 1B*).

Further attempts to understand how *unc-3* might regulate H3K9me have not yet been successful. First, *hpl-1, hpl-2*, and *met-2* gene expression is unaffected in *unc-3* mutants (*Figure 6—figure supplement 1C*). Second, H3K9me3 can be detected in both wildtype and *unc-3* animals by immunostaining (*Figure 6—figure supplement 1D*). A third possibility is that the distribution of H3K9me3 over the genome may be affected in *unc-3* mutant cholinergic MNs. We attempted to test this hypothesis by performing co-localization assays between the *gcy-5 LacO* transgene and H3K9me3, MET-2::mKATE2, or HPL-1/2::mKATE2 in wild-type and *unc-3* deficient MNs. However, these tools were not sufficiently sensitive to answer this question. H3K9me3 localization at the *gcy5* LacO array could not be quantified as the structure of the array was not well preserved after the H3K9me3 staining procedure (Materials and Methods). MET-2 and HPL-1/2 localization could not be quantified because of low expression levels in neurons. Therefore, it remains to be seen if terminal selectors like *unc-3* regulate the correct deposition of H3K9me or if they regulate other functions mediated by *met-2* and HP1 to make non-identity defining genes less accessible.

## Discussion

Differentiated cells are defined by identity-specific transcriptional programs that are stably maintained throughout the life of a cell (*Blau and Baltimore, 1991*; *Deneris and Hobert, 2014*). Here, we provide evidence that in *C. elegans* neurons, identity-defining terminal selectors not only initiate and maintain the expression of effector genes that define a cellular phenotype but they also trigger the restriction of cellular plasticity. We propose that the coordination of identity specification and restriction of plasticity ensures the robustness of the differentiated state, protecting cells from accidental transcription of unwanted genes in response to fluctuations in the environment (*Voss and Hager, 2014*).

### Differentiation and the restriction of cellular plasticity

The loss of developmental plasticity in differentiating cells can be experimentally demonstrated as a cell's inability to respond to ectopically expressed TFs. To assay plasticity, we have ubiquitously expressed a terminal selector TF, CHE-1, at various developmental time points in *C. elegans* and

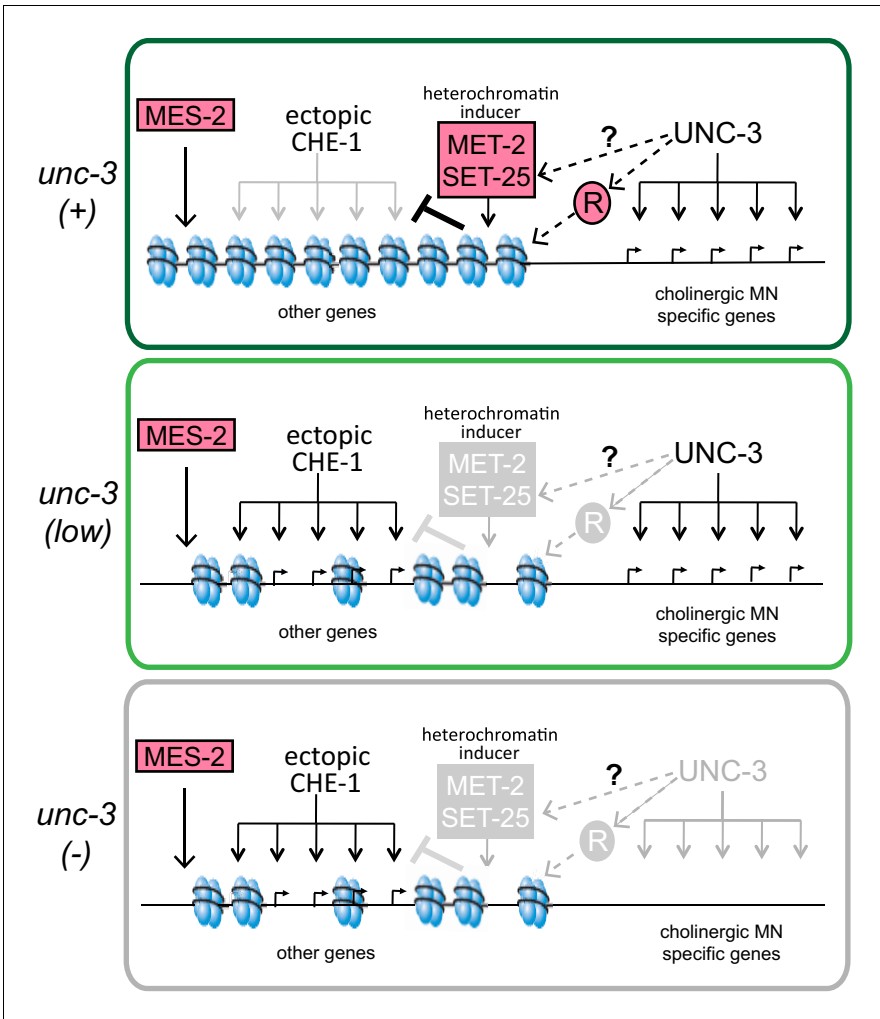

**Figure 7.** A model for the role of terminal selectors in restriction of plasticity. Grey indicates lack of protein activity. In wildtype neurons, *unc-3* activates expression of target genes to specify cholinergic MN identity but it also triggers the restriction of plasticity by activating pathways that repress genes not required for cholinergic MN identity. The repressed genes include ASE effector genes, which cannot be activated by ectopic CHE-1 in wildtype MNs. To restrict plasticity, *unc-3* may regulate the correct distribution of H3K9 methylation through unknown pathways and may also activate additional repressors (R). These repressor mechanisms may then act to 'survey' the genome for genes that have not been activated during identity specification and transform them into a 'non-activatable' stage via H3K9-dependent heterochromatinization. *unc-3* is required to trigger this surveillance mechanism by controlling the function or expression of the genes involved in this process. Independently, *mes-2*-dependent H3K27 methylation also plays a role in gene silencing. Upon lowering levels of *unc-3* activity (*ot837* allele) activation of cholinergic MN genes still takes place, however the *unc-3* dependent pathways that repress other identity genes are not activated, as shown by the ability of CHE-1 to activate its target genes. When *unc-3* is absent, neither specification nor restriction takes place so that cholinergic MN specific genes are not expressed and CHE-1 targets can be activated. A loss of both *unc-3* and *mes-2* results in even more open chromatin, and an increased activation of CHE-1 target genes.

scored its ability to activate fluorescent reporters of target genes. CHE-1 is normally expressed only in 2 ASE sensory neurons where it activates an ASE-specific gene battery. We found that cells that are dividing and beginning to acquire lineage and identity-specific markers are receptive to the activity of ectopic CHE-1. Once cells are postmitotic, they remain receptive for a limited window of time while they are activating terminal identity genes, and then become refractory. A progressive loss of ectopic TF activity in postmitotic cells has also been reported in the mouse brain. Fezf-2, a

master regulatory gene for corticospinal MNs, is capable of activating target genes in embryonic and early postnatal callosal projection neurons and retains its ability to act at postmitotic day 3 and 6. However, this ectopic activity progressively declines and is exhausted by day 21 (*Rouaux and Arlotta, 2013*). These data show that a change occurs in differentiated post-mitotic cells that makes non-identity-specific genes less likely to get activated. In this work, we find that this change is triggered by the resident terminal selector of the differentiated cells in conjunction with downstream and parallel chromatin changes.

We found that the loss or reduction of terminal selector TF activity keeps some neurons in a more plastic state. Five of seven terminal selectors tested in our CHE-1$^{hs}$ assay are required in distinct neuron types to make genes normally expressed in ASE neurons inaccessible to ectopically expressed CHE-1, suggesting that the coordination of differentiation and restriction of cellular plasticity is a commonly occurring phenomenon. The difference between TFs that do and do not function to restrict plasticity is not clear. The two terminal selector-depleted cell types that do not respond to CHE-1 are not lineally more distant from ASE neurons than the five terminal selector mutant cell types that do respond to CHE-1. A comprehensive expression profile of some of these neurons in the presence and absence of the respective TFs may resolve the mechanistic basis for this difference.

In mice, the terminal selector Pax5, which is required for the differentiation of B lymphocytes, also functions to restrict cellular plasticity. Non-terminally differentiated Pro-B cells are responsive to the pluripotency TFs c-Myc, Klf4, Sox2, and Oct4 and can be converted into iPSCs but differentiated B lymphocytes are refractory to the activity of these TFs. Pax5 mutant B lymphocytes, on the other hand, can be converted into iPSCs by the activity of the same TFs (*Hanna et al., 2008*). This suggests that the activity of TFs in restricting plasticity is conserved in other systems. There is, however, a significant difference that makes the function uncovered in this work novel. Pax5(-) B lymphocytes have been shown to retain characteristics of multipotent hematopoietic progenitor Pro-B cells, making it unclear whether it is the transformation into a less differentiated cell type or the lack of Pax5 itself that makes these cells plastic (*Nutt et al., 1999*). In contrast, we find that terminal selector-mutant cells can retain plasticity even if they are not in a progenitor-like state. While terminal selector mutant neurons do not express neuron-type-specific genes, they clearly differentiate into a neuron-like terminal end state. They are postmitotic, still adopt a neuronal morphology, and normally activate expression of pan-neuronal features such as presynaptic vesicle machinery (*Cassata et al., 2000*; *Etchberger et al., 2007*; *Kratsios et al., 2011*; *Stefanakis et al., 2015*). Additionally, *cnd-1*, a homolog of bHLH protein NeuroD, which is expressed transiently during the progenitor-like state in cholinergic MNs, is turned off normally in an *unc-3* mutant (*Hallam et al., 2000*; *Kratsios et al., 2011*). Moreover, mutant neurons of most terminal selectors tested in this assay do not even show a complete loss of expression of terminal identity genes. In *lin-11* mutants, for instance, expression of various terminal identity genes is reduced in ASG, but most tested markers are not completely eliminated (*Sarafi-Reinach et al., 2001*). Similarly, in *ttx-1* mutant AFD neurons, the expression of the glutamate transporter is off in only ~20% of neurons, while in a *ceh-14; ttx-1* double mutant it is off or dim in ~80% of neurons (*Serrano-Saiz et al., 2013*). However, the *gcy-5$^{fosmid}$::gfp* locus is amenable to CHE-1 activation in both the *ttx-1* single and *ttx-1;ceh-14* double mutants to a similar extent. And finally, in the hypomorphic *unc-3::mNG::AID* allele generated in this work, the cholinergic MNs seem to express cholinergic markers and move like wildtype worms suggesting that the differentiation of these cells is minimally affected. Yet, the response to CHE-1 activity in this allele is indistinguishable from the *unc-3* null allele in which MNs are entirely dysfunctional. Hence, the loss of a terminal selector does not simply eliminate the differentiated state, rendering cells more plastic by default. Rather, terminal selectors trigger plasticity restriction by a pathway that can be partially decoupled from identity acquisition.

Earlier acting selector genes of organ identity do not restrict cellular plasticity, as exemplified by the *C. elegans* pharyngeal organ selector *pha-4* (*Yuzyuk et al., 2009*). This is in accordance with these earlier acting selector genes exerting their activity when cells still divide and have not yet acquired a stable and terminal identity.

## Possible mechanism of action of terminal selectors in restricting plasticity

In all terminal selector mutants tested, CHE-1 activates ASE-identity genes in only about 30% of affected neurons. This suggests that restriction of cellular plasticity also requires input from terminal selector-independent factors. In the case of *unc-3* dependent cholinergic neurons, the removal of both an H3K27 methyltransferase, *mes-2*, and an H3K9 methyltransferase, *met-2*, results in the *gcy-5^{prom}::gfp* locus being more accessible to CHE-1. We find that *unc-3* and *mes-2* show an additive phenotype, suggesting that in these neurons, H3K27me3 is one of the terminal selector-independent mechanisms that contribute to restriction of cellular plasticity. On the other hand, a *met-2; unc-3* double mutant phenocopies *unc-3*, suggesting that the incorrect distribution of H3K9 methylation could be a downstream effect of *unc-3* function. This is congruent with the finding that *unc-3* mutants display an expansion of the chromatin structure of a *gcy-5* transgenic reporter.

The role of H3K27me3 in restricting plasticity has been well studied in the *C. elegans* embryo and the germline (*Fakhouri et al., 2010*; *Patel et al., 2012*). However, as overall levels of H3K27me3 do not seem to be affected in somatic cells of L4 stage *mes-2* mutants (*Bender et al., 2004*), this would suggest that the lack or incorrect distribution of H3K27me3 in early stages of larval development is sufficient to interfere with restriction. H3K9me3 has been reported to function in preventing iPSC conversion of fibroblasts (*Becker et al., 2016*). However, in *C. elegans*, somatic developmental defects have not yet been observed in H3K9 methyltransferase mutants (*Zeller et al., 2016*). The current discovery that *met-2* and the homologs of HP1 function to restrict plasticity highlights the utility of TF overexpression assays in uncovering mechanisms of development.

How terminal selectors might modulate chromatin compaction remains unresolved. In cholinergic MNs there is no obvious difference in the gross, overall levels of H3K9me3 between wildtype and *unc-3* mutants. *unc-3* does not seem to be playing a significant role in activating the expression of the *met-2* or *hpl-1/2* genes. Moreover, *unc-3* does not appear to regulate the distribution of H3K9me3 marks indirectly by controlling the deposition of positive marks (H3K4me3 and H3K36me3) at genes that it transcriptionally activates, since we find that the loss of enzymes that generate these marks does not promote cellular plasticity.

Collectively, our data indicate that during identity specification, terminal selectors trigger a 'surveillance mechanism' which silences non-expressed, non-identity-specific genes. Genetic epistasis experiments suggest that this surveillance mechanism is (a) likely to be mediated by *met-2* and *hpl-1/2* and (b) carried out by the correct deposition of H3K9me on genes that are not activated in a given neuron type. This hypothesis can be tested in the future by performing cell-type-specific ChIP-seq experiments to determine the distribution of H3K9me3 in wild-type and terminal selector mutant neurons. How terminal selectors may modulate the deposition of chromatin marks remains to be discovered. One possibility is that terminal selectors directly activate target genes that define a specific neuronal identity while also directly silencing genes that are normally not expressed in the respective neuron type. Although some TFs can act as both activator and repressor depending upon the presence of other co-factors (*Ip, 1995*; *Sakabe et al., 2012*), we disfavor this model because it implies that to be kept in a 'non-activatable' state, all genes would need a unique terminal selector response element for every cell type that they are not expressed in. This mechanism implies an unreasonably complex gene regulatory architecture involving an exceptional number of *cis*-regulatory elements. We rather favor at this point a model in which (a) heterochromatin forms, perhaps by default mechanisms, on loci that are not activated during cellular differentiation and (b) *unc-3* regulates the expression or function of some of the proteins that organize the heterochromatic state (*Figure 7*). These organizers may be broadly expressed heterochromatin regulators that are 'plugged' into cell-specific differentiation program via their activation by cell-specific terminal selectors. Such regulatory linkage would provide a checkpoint to coordinate the induction of cellular differentiation programs with the loss of cellular plasticity.

## Materials and methods

### Worm methods and strains

All strains were maintained using standard protocols. For heatshock experiments, worms were grown at 15°C, heatshocked at 32°C for 30 min, left at 20°C overnight and scored about 24 hr later.

## Strains

OH9846: *otIs305 [hsp16-41$^{prom}$::che-1::: :3XHA::BLRP; rol-6(su1006)]; ntIs1 [gcy-5::gfp; lin-15b(+)]*

OtTi6: *hsp16-41$^{prom}$::che-1::2xFLAG (miniMos single copy insertion)*

OH13139: *otIs587 [gcy-5$^{fosmid}$::sl2::1xnlsgfp; ttx-3$^{prom}$::mcherry]*

OH13102: *otIs586 [gcy-6$^{fosmid}$::sl2::1xnlsgfp; ttx-3$^{prom}$::mcherry]*

OH13984: *ot835 (gcy-5::sl2::1xnls::mNeonGreen)* CRISPR engineered into the genome

OH14049: *otIs629[gcy-7$^{prom}$::tagrfp; ttx-3$^{prom}$::gfp]; ntIs1[gcy-5$^{prom}$::GFP]*

OH13099: *otIs583 [gcy-5$^{prom306bp4XASE}$::gfp; ttx-3$^{prom}$::mcherry]*

OH14218: *otEx6597 [ric-19$^{prom}$::che-1; myo-2$^{prom}$::bfp]*

OH14219: *otEx6598 [unc-47$^{prom}$::che-1; myo-2$^{prom}$::bfp]*

OH11979: *otEx5430 [ift-20$^{prom}$::che-1; rol-6(d)]*

OH8882: *otEx3909 [ceh-36$^{fosmid}$::yfp]*

OH10689: *otIs355 [rab-3$^{prom}$::nlstagrfp]*

OH10598: *otIs348 [unc-47$^{prom}$::mchopti; pha-1]*

OH11124: *otIs388 [eat-4$^{fosmid}$::yfp::h2b]*

OH13988: *ieSi57 [Peft-3::TIR1::mRuby:;unc-54 3'UTR] II; ot837(unc-3::mNeonGreen::AID)* protein tag CRISPR engineered into the genome

OH14021: *ot841 (hpl-1::mKate2)* protein tag CRISPR engineered into the genome

OH14220: *ot860 (hpl-2:: mKate2)* protein tag CRISPR engineered into the genome

OH14221: *ot861 (met-2:: mKate2)* protein tag CRISPR engineered into the genome

OH12738: *otIs545 [gcy-5$^{prom}$::rfp; LacO repeats]; otIs593 [let-858$^{prom}$::gfp::LacI; ttx-3$^{prom}$::mcherry]*

OH11744: *otIs445 [ace-2$^{prom}$::rfp; LacO repeats;] gwIs39 [baf-1$^{prom}$::gfp::lacI; vit-5$^{prom}$::gfp]*

## Mutant alleles

*unc-3(e151), ceh-14(ch3), lin-11(n389), unc-30(e191), unc-42(e419), ets-5(tm1734), ttx-1(p797), met-2 (ok2307), set-25(n5021), hpl-1(tm1489), hpl-2(n4317), unc-55(e402), unc-4(e120), mab-9(gk396730), SS186: (mes-2(bn11) unc-4(e120)/mnC1 dpy-10(e128) unc-52(e444)II)*

## MiniMos

MiniMos was performed exactly as presented in *Frøkjær-Jensen et al. (2014)*. *hsp16.41$^{prom}$::che-1::2Xflag* was cloned into the miniMOS vector pCFJ910 and integrated into the genome. Three independent insertions were obtained of which OTti6 was used for all experiments.

## CRISPR alleles

CRISPR engineering was performed exactly as described in *Dickinson et al. (2015)*. For *gcy-5*, SL2::1XNLS::mNeonGreen was inserted after the stop codon of *gcy-5*. For *unc-3*, mNeonGreen::AID tag was inserted into the C-terminus of the protein just before the stop codon. For *hpl-1*, mKate2 was inserted into the protein after the 12th amino acid. This tag was based on a published hpl-1 protein fusion (*Couteau et al., 2002*). For *hpl-2*, mKate2 was inserted into the protein after the 96th amino acid. This tag was based on a published *hpl-1* protein fusion (*Couteau et al., 2002*). There are two isoforms of *hpl-2*, both are tagged in this allele. For *met-2*, mKate2 was added to the C-terminus of the protein followed by a small protein bridge present in the plasmids published in *Dickinson et al. (2015)*. The start codon of *met-2* was kept intact. The MET-2 protein seems to localize inside the nucleus in opposition to a previously published expression pattern (*Towbin et al., 2012*).

## Fosmid recombineering

Fosmid recombineering to create *otIs586* and *otIs587* was performed as in *Tursun et al. (2009)*. The recombineered fosmids were linearized and injected into hermaphrodites as complex arrays with 15 ng/µl of the linearized fosmid, 3 ng/µl of *ttx-3$^{prom}$::mcherry* used as a co-injection marker and 100 ng/µl of digested bacterial genomic DNA. Extrachromosomal lines acquired after injection were integrated into the genome using gamma irradiation.

## LacO arrays

To create the *gcy-5* LacO arrays, 10 ng/µl of PCR product containing *gcy-5^prom::rfp* were injected with 2 ng/µl of LacO repeats acquired by digesting out the10 kb Sph1/Kpn1 fragment from lacO multimeric plasmid pSV2-dhfr- 8.23, and 100 ng/µl of digested bacterial genomic DNA were injected into GW396 young adult hermaphrodites. Extrachromosomal arrays that formed nuclear LacI::GFP spots were chosen for integration with gamma irradiation. Integrants that showed nuclear spots were then chosen.

The *ace-2* LacO arrays were similarly built. The injection mix contained: 10 ng/µl of *ace-2^prom::rfp* PCR, 2 ng/µl of LacO repeats, 2 ng/µl of *elt-2::nlsdsred*, and 86 ng/µl of digested bacterial genomic DNA. Extrachromosomal and integrant lines were chosen as with *gcy-5*.

## DiI filling

The amphid sensory neurons of worms can uptake fluorescent dyes (*Shaham, 2006*). DiI labeling was performed by washing worms into 1 mL of M9 butter containing 1:500 dilution of DiI. Worms were left to incubate with the dye for 1 hr at room temperature, washed three times with M9 and placed on NGM plates with *E. coli* for >1 hr to recover before imaging.

## Antibody staining

A modified version of the protocol published in *Gendrel et al. (2016)* was used for GFP and H3K27me3 staining. Worms were fixed in 4% paraformaldehyde/PBS for 15 min at 4°C, washed three times in 0.5% triton/PBS, rocked gently in 5% $\beta$-mercaptoethanol/1% Triton X-100/0.1 M Tris-HCl(pH 7.5) for 18 hr at 37°C, washed four times in 1% Triton X-100/0.1 M Tris-HCl(pH7.5) and once in 1 mM $CaCl_2$/1% Triton X-100/0.1 M Tris-HCl(pH7.5). A worm pellet of 20–50 µL was then shaken in 1 mg/mL of collagenase type IV (C5138, Sigma)/1 mM $CaCl_2$/1% Triton X-100/0.1 M Tris-HCl (pH7.5) at 37°C for 30 min. The worms were then washed three times in PBS/0.5% Triton X-100, blocked, and stained. Blocking was done in PBS/0.2% gelatin/0.25% Triton for 30 min at room temperature. Antibodies were diluted in PBS/0.1% gelatin/0.25% Triton. Primary antibody was left on overnight at 4°C or 1 hr at room temperature and secondary antibody was applied for 3 hr at room temperature. After washing off the secondary antibody, worms were incubated with DAPI for 15 min, washed again and mounted on glass slides. Primary antibodies used were: H3K27me3 (Millipore, 1:500) and GFP (Invitrogen, 1:1000 dilution). All secondary antibodies were Alexa Flour dyes used at 1:1000 dilution.

For H3K9me3 staining, we found that anti-H3K9me3 (Abcam,1:500 dilution) showed non-specific staining as observed by the presence of staining in *met-2; set-25* embryos and L1s, which have been shown by Mass Spectroscopy to not have any H3K9me3 (*Towbin et al., 2012*). We found the Kimura H3K9me3 antibody, CMA318 purchased from MBL International, to be specific. For this antibody, the paraformaldehyde fixation described above did not work in somatic cells and a freeze crack antibody staining protocol on whole worms was used (*Duerr, 2006*). Worms were washed, suspended in 0.025% glutaraldehyde/$ddH_2$0 and spread out in between two frost-resistant glass slides. These slides were frozen on dry ice and cracked open to break the cuticle of the animals. Freeze cracked worms were fixed in ice-cold acetone for 5 min and then in ice-cold methanol for 5 min. The worms were then washed off the slides in PBS, blocked, stained, and mounted as above.

## Microscopy

µManager and Zen were used for image acquisition and processing (*Edelstein et al., 2010*).

## Single molecule fluorescent in situ hybridization (smFISH)

smFISH was performed using Custom Stellaris FISH probes, purchased from Biosearch Technologies. Staining was performed according to the manufacturer's protocol.

## Acknowledgements

We thank Dylan Rahe for extensive discussions; Qi Chen for expert assistance in strain generation; Marie Gendrel for optimizing the immunostaining protocol; Baris Tursun for early guidance in the project; CGC for strains; members of the Hobert Lab for comments on the manuscript. TP was

funded by a National Science Foundation GRFP fellowship. OH is an nvestigator of the Howard Hughes Medical Institute. This work was also supported by the National Institutes of Health (R21 NS076191-01).

## Additional information

### Competing interests

OH: Reviewing Editor, *eLife*. The other author declares that no competing interests exist.

### Funding

| Funder | Author |
| --- | --- |
| Howard Hughes Medical Institute | Oliver Hobert |
| National Institute of Neurological Disorders and Stroke | Oliver Hobert |

The funders had no role in study design, data collection and interpretation, or the decision to submit the work for publication.

### Author contributions

TP, Conceptualization, Formal analysis, Investigation, Visualization, Methodology, Writing—original draft, Writing—review and editing; OH, Conceptualization, Supervision, Funding acquisition, Project administration, Writing—review and editing

### Author ORCIDs

Tulsi Patel, http://orcid.org/0000-0002-3635-3567
Oliver Hobert, http://orcid.org/0000-0002-7634-2854

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
