## [Decision Letter]

Thank you for submitting your article "Coordinated control of terminal differentiation and restriction of cellular plasticity" for consideration by *eLife*. Your article has been favorably evaluated by K VijayRaghavan (Senior Editor) and three reviewers, one of whom, Kang Shen (Reviewer #1), is a member of our Board of Reviewing Editors.

The reviewers have discussed the reviews with one another and the Reviewing Editor has drafted this decision to help you prepare a revised submission.

Summary:

Patel and Hobert tested the relationship between cellular differentiation and cellular plasticity. They performed thorough and elegant experiments using the *C. elegans* TF *che-1*. They found that expression of *che-1* induces ectopic expression of neuronal specific genes, but only in young animals. This restriction is lost in multiple but not all selector mutants. They further showed that chromatin compaction is affected in the selector mutants. While the direct link between the selector genes and chromatin state is not clear, this exciting manuscript produced important insights into the mechanistic link between differentiation and restriction of plasticity.

Essential revisions:

1) In Figure 1, the authors used an artificial construct with 4XASE-GFP to show that adult heatshock of *che-1* is still able to produce transcriptionally active CHE-1. However, this assay uses an overexpressed, 4X transgene. Why not just tag the *hs-che-1* with mcherry? That should provide a direct measure of the level of expression achieved by heatshocking at different stages. Since this is a very important manipulation for the paper, I suggest that the authors do this experiment.

2) What could the difference between the five terminal selectors that are required for restricting plasticity and the two that are not? Is it because there's a fate transformation in the latter group, so that another terminal selector is expressed? Or perhaps the degree of lineage separation from ASE?

3) I find the result intriguing where ectopic expression of CHE-1 in hypoderm and muscle in the embryo could induce *gcy-5* expression. These tissues are separated from neuronal fate very early in embryogenesis. Supposedly, layers of fate restriction would have piled on by the time of terminal differentiation (e.g., the studies cited in the third paragraph of Introduction). Would this imply that terminal differentiation triggers a different kind and qualitatively stronger restriction? If so, it would enhance the interpretation of the main results: the requirement of the examined terminal selectors is not simply a result of neuronal subtypes being specified late from a generic neuronal fate. If the authors so wish, it would be interesting to see if non-neuronal genes (e.g., muscle) can be expressed easier in *unc-3* mutants than in the WT.

4) The last paragraph of the Discussion considers possible mechanisms to explain this conundrum but the suggested alternatives seem baroque and difficult to follow. Perhaps this section can be clarified by careful editing.

5) Of greater concern is the absence of direct evidence that a "terminal selector" transcription factor (UNC-3 in this case) prevents ectopic gene induction by triggering the local application of H3K9 methylation marks to silence transcription. Although the authors perform a convincing genetic epistasis experiment to confirm that the H3K9 modifying enzyme met-2 functions in the *unc-3* pathway to restrict "cellular plasticity," this experiment does not rule out the possibility that the *met-2* gene product could also affect gene expression by other means. Moreover, immunostaining did not detect a quantitative difference in H3K9me levels in wildtype vs. *unc-3* mutant neurons. The authors suggest that *unc-3* may affect the distribution of H3K9me marks but not the overall level. This explanation seems plausible but is not directly tested. What is needed here is an experiment to ask if H3K9me marks are specifically applied to off target genes (i.e., not directly *unc-3* regulated) in neurons that express UNC-3. A ChIP-seq experiment that specifically interrogates genes in the UNC-3-expressing neurons could resolve this question but this set up may not be technically feasible. Have the authors considered other alternative strategies for tackling this question? At the very least, this technical caveat should be directly addressed in the text.

---

## [Author Response]

*Essential revisions:*

*1) In Figure 1, the authors used an artificial construct with 4XASE-GFP to show that adult heatshock of che-1 is still able to produce transcriptionally active CHE-1. However, this assay uses an overexpressed, 4X transgene. Why not just tag the hs-che-1 with mcherry? That should provide a direct measure of the level of expression achieved by heatshocking at different stages. Since this is a very important manipulation for the paper, I suggest that the authors do this experiment.*

The heatshock *che-1* construct used in this assay is HA-tagged, and therefore can be detected by staining. We have confirmed the presence of the heatshock induced CHE-1 by both staining for HA and by performing smFISH of *che-1*mRNA. These figures have now been added in Figure 1—figure supplement 2. The utility of the 4XASE-GFP is in showing not only that the CHE-1 protein is expressed in response to heatshock at all stages, but also in confirming that the expressed CHE-1 protein is active as a transcription factor. This clarification has now been added to the text.

*2) What could the difference between the five terminal selectors that are required for restricting plasticity and the two that are not? Is it because there's a fate transformation in the latter group, so that another terminal selector is expressed? Or perhaps the degree of lineage separation from ASE?*

This is indeed interesting to consider – there is no evidence, as far as we know, that the GABA-ergic neurons and the ASH differentiate into another cell fate in *unc-30* and *unc-42* mutants respectively (in which *che-1* is not able to reprogram identity). Additionally, lineage separation also does not explain the discrepancy as *unc-3* dependent cholinergic motor neurons are lineally very closely related to the GABA-ergic motor neurons and equally distant from *che-1* dependent ASE neurons. Another explanation for the difference could reside in the properties of the transcription factors themselves – however, we did not find any differences in the protein domains in the transcription factors that do and do not regulate plasticity. A broader understanding of the functions of these transcription factors, such an assessment of gene expression in wt vs. mutant neurons, might be required to answer this question. There is evidence that both *unc-42* and *unc-30* work with co-factors to specify their respective neuronal fates. It is also possible that these co-factor transcription factors control cellular plasticity in these cells. These points are now briefly raised in the Discussion.

*3) I find the result intriguing where ectopic expression of CHE-1 in hypoderm and muscle in the embryo could induce gcy-5 expression. These tissues are separated from neuronal fate very early in embryogenesis. Supposedly, layers of fate restriction would have piled on by the time of terminal differentiation (e.g., the studies cited in the third paragraph of Introduction). Would this imply that terminal differentiation triggers a different kind and qualitatively stronger restriction? If so, it would enhance the interpretation of the main results: the requirement of the examined terminal selectors is not simply a result of neuronal subtypes being specified late from a generic neuronal fate. If the authors so wish, it would be interesting to see if non-neuronal genes (e.g., muscle) can be expressed easier in unc-3 mutants than in the WT.*

We did try this experiment, i.e., we expressed *hlh-1* in wt and *unc-3* mutants. However, we did not see the ectopic induction of an *unc-97* reporter in either wt or *unc-3* mutants at any larval stages. Based on these results alone though, we cannot conclude that in *unc-3* mutant neurons muscle genes are not accessible for activation. The inability of *hlh-1* to activate *unc-97* might simply suggest that *hlh-1* is insufficient for the activation of *unc-97* and may require co-factors. There is evidence that *hlh-1* works together with *unc-120* and *hnd-1* to activate muscle genes, supporting the hypothesis that it may not be sufficient in larval somatic tissues to activate *unc-97*. This would be analogous to the inability of *che-1* to activate ASEL genes *gcy-6* and *gcy-7* in all somatic larval tissues. We think that this caveat makes the results from this experiment inconclusive and therefore did not include them.

*4) The last paragraph of the Discussion considers possible mechanisms to explain this conundrum but the suggested alternatives seem baroque and difficult to follow. Perhaps this section can be clarified by careful editing.*

This paragraph has now been edited.

*5) Of greater concern is the absence of direct evidence that a "terminal selector" transcription factor (UNC-3 in this case) prevents ectopic gene induction by triggering the local application of H3K9 methylation marks to silence transcription. Although the authors perform a convincing genetic epistasis experiment to confirm that the H3K9 modifying enzyme met-2 functions in the unc-3 pathway to restrict "cellular plasticity," this experiment does not rule out the possibility that the met-2 gene product could also affect gene expression by other means. Moreover, immunostaining did not detect a quantitative difference in H3K9me levels in wildtype vs. unc-3 mutant neurons. The authors suggest that unc-3 may affect the distribution of H3K9me marks but not the overall level. This explanation seems plausible but is not directly tested. What is needed here is an experiment to ask if H3K9me marks are specifically applied to off target genes (i.e., not directly unc-3 regulated) in neurons that express UNC-3. A ChIP-seq experiment that specifically interrogates genes in the UNC-3-expressing neurons could resolve this question but this set up may not be technically feasible. Have the authors considered other alternative strategies for tackling this question? At the very least, this technical caveat should be directly addressed in the text.*

Yes, a cell-specific ChIP-seq would be the ideal experiment, however it is technically challenging. In the absence of this, we did try to use the *gcy-5 lacI/O* array to address this question. Specifically, we asked if the localization of H3K9me3, HPL-2, HPL-1 or MET-2 changes at the *gcy-5* LacI/LacO transgenic locus in wt vs. *unc-3* mutant neurons. This, however, also proved technically unreliable for the following reasons: For detecting H3K9me3, we performed immunostaining and tried to ask if the colocalization between the LACI::GFP (bound at the *gcy-5 LacO* locus) and H3K9me3 changes in wt and *unc-3* mutants. However, this staining procedure required an acetone/methanol fixation, which did not preserve the chromatin architecture (as was obvious by comparing the *ace-2 LacO* arrays in wt and *unc-3* mutants after acetone/methanol fixation vs. live imaging or paraformaldehyde fixation), and co-staining experiments were therefore unreliable. We also tried to measure the co-localization of HPL-1/2 and MET-2 using the mKATE-2 tagged CRISPR alleles at the *gcy-5 LacI/O* locus, but could not reliably image fluorescence because of low expression levels in neurons. The fact that the manuscript does not show the functional relationship between *unc-3* and *met-2* mediated chromatin effects has now been further clarified by the addition of these attempted experiments and an edited Discussion.